# Turning the Tables: Enabling Backward Transfer via Causal-Aware LoRA in Continual Learning

**Chaoyang Li**[1,2]  **Runze Ye**[1]  **Jianyang Qin**[1]  **Jinhao Cui**[1]

**Lingzhi Wang**[1]  **Ning Hu**[2]

**Qing Liao**[1,2]*

[1]Harbin Institute of Technology, Shenzhen, China
[2]Peng Cheng Laboratory, Shenzhen, China
`{22b951022, 24S151081, 22b351005, cuijinhao}@stu.hit.edu.cn,`
`{hun}@pcl.ac.cn,{wanglingzhi, liaoqing}@hit.edu.cn`

## Abstract

Current parameter-efficient fine-tuning (PEFT) methods have shown superior performance in continual learning. However, most existing PEFT-based methods focus on mitigating catastrophic forgetting by limiting modifications to the old task model caused by new tasks. This hinders backward knowledge transfer, as when new tasks have a strong positive correlation with old tasks, appropriately training on new tasks can transfer beneficial knowledge to old tasks. Critically, achieving backward knowledge transfer faces two fundamental challenges: (1) some parameters may be ineffective on task performance, which constrains the task solution space and model capacity; (2) since old task data are inaccessible, modeling task correlation via shared data is infeasible. To address these challenges, we propose CaLoRA, a novel **c**ausal-**a**ware **lo**w-**r**ank **a**daptation framework that is the first PEFT-based continual learning work with backward knowledge transfer. Specifically, we first propose **pa**rameter-level **c**ounterfactual **a**ttribution (PaCA) that estimates the causal effect of LoRA parameters via counterfactual reasoning, identifying effective parameters from a causal view. Second, we propose **c**ross-**ta**sk **g**radient **a**daptation (CaGA) to quantify task correlation by gradient projection and evaluate task affinity based on gradient similarity. By incorporating causal effect, task correlation, and affinity, CaGA adaptively adjusts task gradients, facilitating backward knowledge transfer without relying on data replay. Extensive experiments across multiple benchmarks and continual learning settings show that CaLoRA outperforms state-of-the-art methods. In particular, CaLoRA better mitigates catastrophic forgetting by enabling positive backward knowledge transfer.

## 1 Introduction

Recently, parameter-efficient fine-tuning (PEFT) [1–7] has emerged as a promising technology for continual learning (CL), offering a feasible solution to maintain performance on old tasks and improving learning capabilities for new ones [8]. Among the most widely adopted techniques include prompt-tuning [1–4], adapter-tuning [9–11], and low-rank adaptation (LoRA) [6, 7, 12, 13], all of which primarily address catastrophic forgetting [1, 6, 9, 12]. These methods employ two fundamental

---

*Corresponding author.

39th Conference on Neural Information Processing Systems (NeurIPS 2025).

Table 1: Comparison of PEFT-based continual learning methods.

| Method | Data Replay-Free | Efficient Inference | Backward Transfer |
|---|---|---|---|
| CodaPrompt [20](CVPR'23) | ✓ | ✗ | ✗ |
| HidePrompt [21](NeurIPS'23) | ✗ | ✗ | ✗ |
| InfLoRA [6] (CVPR'24) | ✓ | ✓ | ✗ |
| SAPT [13] (ACL'24) | ✗ | ✓ | ✗ |
| SD-LoRA [7] (ICLR'25) | ✓ | ✓ | ✗ |
| CaLoRA(ours) | ✓ | ✓ | ✓ |

strategies: selectively incorporating old task parameters during new task learning [1, 3, 7, 9, 13], or ensuring orthogonal updates to new task parameters relative to old ones [6, 12].

Although most existing continual learning methods based on PEFT effectively mitigate catastrophic forgetting [1, 3, 6, 9], the parameter-freezing strategy inherently limits backward knowledge transfer by treating updates for new tasks as potential interference with old tasks [6, 12, 14]. The learning parameters of new tasks do not necessarily conflict with those of old tasks [15, 16]. When the new task is positively correlated with old ones, it can facilitate backward knowledge transfer by contributing beneficial information to old tasks [16]. However, existing continual learning methods based on PEFT have not yet explored the backward knowledge transfer, as shown in the Table. 1.

This motivates us to propose a novel continual learning approach based on PEFT that addresses forgetting and enhances old tasks through effective backward knowledge transfer. To achieve this, it is essential to understand when and how backward knowledge transfer occurs. When a new task is strongly correlated with old tasks, it creates favorable conditions for knowledge transfer [15, 16]. In this case, if this transfer is positive (i.e., the new task can transfer beneficial knowledge to old tasks), we can promote backward knowledge transfer by optimizing parameters beneficial to old tasks during the training of the new task. Conversely, if the transfer is negative (i.e., the new task interferes with old tasks), it is necessary to protect old task knowledge to mitigate catastrophic forgetting.

However, realizing the above objective faces two key challenges:(1) Although existing PEFT methods can achieve performance comparable to full-parameter training by tuning only a few parameters, some parameters may be ineffective in enhancing performance [17–19]. Moreover, as the number of tasks increases, the parameter solution space for new tasks becomes increasingly constrained by old tasks, leading to worse performance [15]. (2) Since data from old tasks is not replayable, traditional methods for modeling task correlation via shared data are infeasible [7, 15, 16].

To address these challenges, we propose CaLoRA, a novel **c**ausal-**a**ware **lo**w-**r**ank **a**daptation framework, which is the first PEFT-based continual learning work to enable backward knowledge transfer. Specifically, to tackle the first challenge, we propose the **pa**rameter-level **c**ounterfactual **a**ttribution (PaCA) method that estimates the causal effect of LoRA parameters via counterfactual reasoning. PaCA enables the model to identify effective parameters with strong causal effects for the current task while reducing the influence of ineffective ones, enhancing its ability to learn future tasks. To address the second challenge, we propose the **c**ross-t**a**sk **g**radient **a**daptation (CaGA) method, which measures task correlation by gradient projection and computes task affinity based on gradient similarity in the task space. By jointly leveraging causal effect, task correlation, and affinity, CaGA adaptively adjusts task gradients, enabling effective backward knowledge transfer without data replay. The key contributions of this paper are summarized as follows:

- We propose CaLoRA, a novel **c**ausal-**a**ware **lo**w-**r**ank **a**daptation framework, which is the first PEFT-based continual learning work to mitigate catastrophic forgetting more effectively by actively enabling backward knowledge transfer.

- We propose parameter-level counterfactual attribution to identify effective parameters via counterfactual reasoning, reducing the influence of ineffective parameters in current tasks and expanding the capacity for future tasks.

- We propose cross-task gradient adaptation that adaptively adjusts task gradients based on causal effect, task correlation, and affinity, enabling backward knowledge transfer without data replay.

- Extensive experiments on multiple benchmarks under various task settings demonstrate that CaLoRA consistently outperforms existing state-of-the-art methods in continual learning.

## 2 Related Work

### 2.1 Parameter-Efficient Fine-Tuning.

Parameter-efficient fine-tuning (PEFT) methods involve freezing a pre-trained model and introducing fewer parameters than full fine-tuning, often achieving comparable or even superior performance [22, 23]. For instance, the Adapter[9–11, 18, 24] incorporates small modules at various layers of the pretrained model, tuning only these additional modules for task adaptation. Prompt-tuning [4, 25] and prefix-tuning [26] introduce learnable tokens into the input for the pretrained model, adjusting only these tokens to fit specific tasks. Low-rank adaptation (LoRA) [19, 22, 27] reparameterizes pre-trained weights with two low-rank matrices, tuning solely these matrices for downstream adaptation.

### 2.2 Continual Learning.

Continual learning focuses on sequentially acquiring knowledge from new tasks while maintaining performance on old ones [28, 29]. Existing approaches primarily fall into three categories: replay-based, regularization-based, and expansion-based methods. Replay-based methods [30–34] mitigate forgetting by storing and reusing samples from old tasks. Regularization-based techniques [35–37] constrain updates to parameters critical for old tasks through designing a penalty term. Expansion-based approaches [38, 39] dynamically adjust model capacity to accommodate new tasks while preserving learned knowledge.

Recently, PEFT has shown effectiveness in continual learning by mitigating catastrophic forgetting, including prompt-tuning, adapter-tuning, and LoRA. Specifically, prompt-tuning methods [1–3] introduce learnable prompt tokens into transformer layers. For example, L2P [1], DualPrompt [40], and CODA-Prompt [20] integrate Vision Transformers (ViTs) with learnable embeddings with prompts, thereby improving knowledge retention as new tasks are learned. Building on these, HiDe-Prompt [21] further stores old task samples to enhance performance. Adapter-tuning methods [10, 11] often insert lightweight learnable modules into transformer layers, like C-ADA [11] designs a parameter-extensible continual adapter layer in the pre-trained model. LoRA [12, 22] updates the pre-trained weights with low-rank matrices. For instance, InfLoRA [6] prevents interference between tasks by constraining new task updates to the orthogonal space of old tasks. SD-LoRA [7] decouples the learning of the magnitude and direction of LoRA components. Moreover, SAPT [13] aligns the PEFT block (including prompt and LoRA) via a shared attentive module and replays old task samples to improve performance. Despite these advances, existing PEFT-based continual learning methods fail to simultaneously satisfy all desirable properties outlined in Table 1. To bridge this gap, we propose CaLoRA, a novel **c**ausal-**a**ware **lo**w-**r**ank **a**daptation framework that enables backward knowledge transfer without data replay.

## 3 Methodology

Fig. 1 illustrates the CaLoRA framework, which incorporates **pa**rameter-level **c**ounterfactual **a**ttribution (PaCA) and **c**ross-t**a**sk **g**radient **a**daptation (CaGA). First, PaCA estimates causal effects of LoRA parameters via counterfactual reasoning, which identifies effective parameters from a causal view. Second, CaGA measures task correlation and affinity to determine conditions for knowledge transfer. CaGA further adapts gradients based on causal effects, task correlation, and affinity, thereby promoting beneficial knowledge transfer. More details are in subsections 3.2 and 3.3.

### 3.1 Preliminaries

**Problem Definition.** Continual learning tackles the challenge of sequentially learning from multiple tasks, where each task $\mathcal{T}_t$ in the sequence $\{\mathcal{T}_1, \ldots, \mathcal{T}_T\}$ is associated with a distinct dataset $\mathcal{D}_t = \{(x_t^j, y_t^j)\}_{j=1}^{|\mathcal{D}_t|}$. Here, $x_t^j$ denotes an input sample and $y_t^j$ its corresponding label, with $|\mathcal{D}_t|$ representing the dataset size. The model $f_\Theta$, parameterized by $\Theta$, is trained exclusively on $\mathcal{D}_t$ when learning task $\mathcal{T}_t$. The training objective for the $t$-th task is defined as (1):

$$\mathcal{L}(\mathcal{D}_t; \Theta) = \frac{1}{|\mathcal{D}_t|} \sum_{j=1}^{|\mathcal{D}_t|} \mathcal{L}\left(f_\Theta(x_t^j), y_t^j\right), \tag{1}$$

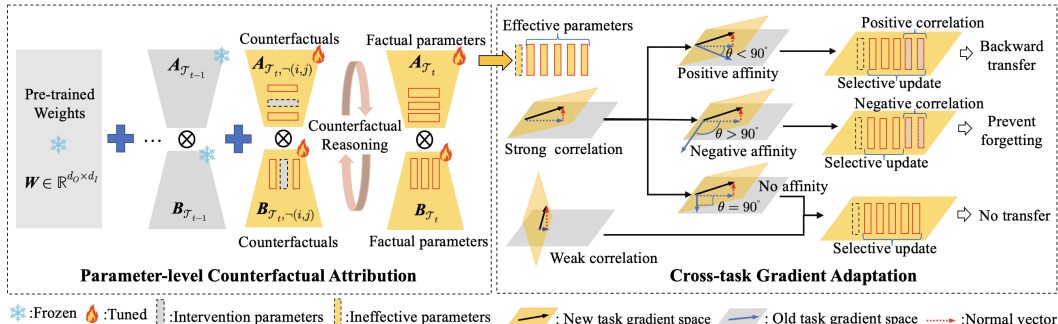

Figure 1: The CaLoRA framework consists of two components. First, Parameter-level Counterfactual Attribution estimates causal effects of parameters via counterfactual reasoning, identifying effective parameters from a causal view. Second, Cross-task Gradient Adaptation assesses task correlation and affinity to determine conditions for effective knowledge transfer. It further adapts gradients based on causal effects, task correlation, and affinity, thereby promoting backward knowledge transfer.

where $\mathcal{L}$ denotes the loss function. The ultimate goal in continual learning is to train the model $f_\Theta$ to ensure good performance not only on the new task $\mathcal{T}_t$ but also on all old tasks $\{\mathcal{T}_1, \ldots, \mathcal{T}_{t-1}\}$.

**Revisit Low-Rank Adaptation.** LoRA [22] aims to constrain parameter updates to a low-rank subspace during fine-tuning. Given a pre-trained weight matrix $\boldsymbol{W} \in \mathbb{R}^{d_O \times d_I}$ for a specific pretrained layer with input dimension $d_I$ and output dimension $d_O$, LoRA re-parameterizes it by introducing two low-rank matrices: $\boldsymbol{A} \in \mathbb{R}^{r \times d_I}$ and $\boldsymbol{B} \in \mathbb{R}^{d_O \times r}$, where $r \ll \min(d_I, d_O)$. Here, $\boldsymbol{A}$ reduces the dimensionality, and $\boldsymbol{B}$ restores it. The modified forward pass becomes $\boldsymbol{z} = \boldsymbol{W}\boldsymbol{h} + \boldsymbol{B}\boldsymbol{A}\boldsymbol{h}$, where $\boldsymbol{h}$ is the input and $\boldsymbol{z}$ is the output. LoRA initializes $\boldsymbol{B}$ to zeros and $\boldsymbol{A}$ with a Gaussian distribution, keeping $\boldsymbol{W}$ fixed while fine-tuning only $\boldsymbol{A}$ and $\boldsymbol{B}$. To achieve the goal of continual learning, given the task sequence $\{\mathcal{T}_1, \ldots, \mathcal{T}_T\}$, each task has two trainable matrices. The output of the $t$-th task at the specific layer is modified as Eq. (2). The LoRA of the $t$-th task updates parameter matrices $\boldsymbol{A}_{\mathcal{T}_t}$ and $\boldsymbol{B}_{\mathcal{T}_t}$ with gradient (i.e., $\boldsymbol{G}_{\mathcal{T}_t} = \nabla_{\boldsymbol{W}_{\mathcal{T}_t}} \mathcal{L}(\mathcal{D}_{\mathcal{T}_t}; \boldsymbol{W}_{\mathcal{T}_t})$) for every step $s$ by Eq.(3).

$$\boldsymbol{z}_{\mathcal{T}_t} = \boldsymbol{W}_{\mathcal{T}_t}\boldsymbol{h} = \boldsymbol{W}\boldsymbol{h} + \sum_{k=1}^{t} \boldsymbol{B}_{\mathcal{T}_k}\boldsymbol{A}_{\mathcal{T}_k}\boldsymbol{h} = \boldsymbol{W}_{\mathcal{T}_{t-1}}\boldsymbol{h} + \boldsymbol{B}_{\mathcal{T}_t}\boldsymbol{A}_{\mathcal{T}_t}\boldsymbol{h}, \tag{2}$$

$$\boldsymbol{A}_{\mathcal{T}_t,(s+1)} \leftarrow \boldsymbol{A}_{\mathcal{T}_t,(s)} - \eta \boldsymbol{B}_{\mathcal{T}_t,(s)}^\top \boldsymbol{G}_{\mathcal{T}_t}, \quad \boldsymbol{B}_{\mathcal{T}_t,(s+1)} \leftarrow \boldsymbol{B}_{\mathcal{T}_t,(s)} - \eta \boldsymbol{G}_{\mathcal{T}_t} \boldsymbol{A}_{\mathcal{T}_t,(s)}^\top. \tag{3}$$

### 3.2 Parameter-level Counterfactual Attribution

Although LoRA constrains parameter updates to a low-rank subspace, existing studies have demonstrated that such low-rank constraints do not guarantee that all parameters are effective in improving task performance [17–19]. In continual learning scenarios, this limitation becomes particularly challenging because new task parameters are typically constrained (e.g., via gradient orthogonality [6, 12, 14]) to protect old tasks. It is worth noting that when ineffective parameters from old tasks are used to constrain new tasks, it not only fails to alleviate catastrophic forgetting but may also harm new task performance [15]. To address this, we propose a **pa**rameter-level **c**ounterfactual **a**ttribution (PaCA) method that uses counterfactual reasoning to assess the causal effects of low-rank parameters, identifying the effective parameters with strong causal effect.

Specifically, to quantify the causal effect of an individual parameter $\boldsymbol{W}_{\mathcal{T}_t,(i,j)}$ in the parameter matrix $\boldsymbol{W}_{\mathcal{T}_t} \in \mathbb{R}^{d_O \times d_I}$, we first perform a causal intervention by setting the parameter to zero [41–43], generating the counterfactual weight matrix $\boldsymbol{W}_{\mathcal{T}_t,\neg(i,j)}$. The individual causal effect is then defined as the change in loss when a parameter is included versus excluded [44–46], as shown in Eq. (4):

$$\boldsymbol{E}_{\mathcal{T}_t,(i,j)} = \mathcal{L}(\mathcal{D}_t; do(\boldsymbol{W}_{\mathcal{T}_t,(i,j)} = 0)) - \mathcal{L}(\mathcal{D}_t; \boldsymbol{W}_{\mathcal{T}_t}) = \mathcal{L}(\mathcal{D}_t; \boldsymbol{W}_{\mathcal{T}_t,\neg(i,j)}) - \mathcal{L}(\mathcal{D}_t; \boldsymbol{W}_{\mathcal{T}_t}), \tag{4}$$

where $do(\boldsymbol{W}_{\mathcal{T}_t,(i,j)} = 0)$ refers to a causal intervention operation that sets $\boldsymbol{W}_{\mathcal{T}_t,(i,j)}$ to zero. However, directly computing the causal effect by intervening on each parameter is computationally prohibitive. To address this, we approximate the counterfactual loss $\mathcal{L}(\mathcal{D}_t; \boldsymbol{W}_{\mathcal{T}_t,\neg(i,j)})$ using a second-order Taylor

expansion [47] around the original weight matrix $\boldsymbol{W}_{\mathcal{T}_t}$. This leads to the quadratic approximation shown in Eq. (5):

$$\boldsymbol{E}_{\mathcal{T}_t,(i,j)} \approx \mathcal{L}(\mathcal{D}_t; \boldsymbol{W}_{\mathcal{T}_t}) - \frac{\partial \mathcal{L}(\mathcal{D}_t; \boldsymbol{W}_{\mathcal{T}_t})}{\partial \boldsymbol{W}_{\mathcal{T}_t,(i,j)}} \Delta \boldsymbol{W}_{\mathcal{T}_t,(i,j)} + \frac{1}{2} \frac{\partial^2 \mathcal{L}(\mathcal{D}_t; W_t)}{\partial \boldsymbol{W}_{\mathcal{T}_t,(i,j)}^2} \Delta \boldsymbol{W}_{\mathcal{T}_t,(i,j)}^2 - \mathcal{L}(\mathcal{D}_t; \boldsymbol{W}_{\mathcal{T}_t})$$

$$= -\frac{\partial \mathcal{L}(\mathcal{D}_t; \boldsymbol{W}_{\mathcal{T}_t})}{\partial \boldsymbol{W}_{\mathcal{T}_t,(i,j)}} \Delta \boldsymbol{W}_{\mathcal{T}_t,(i,j)} + \frac{1}{2} \frac{\partial^2 \mathcal{L}(\mathcal{D}_t; \boldsymbol{W}_{\mathcal{T}_t})}{\partial \boldsymbol{W}_{\mathcal{T}_t,(i,j)}^2} \Delta \boldsymbol{W}_{\mathcal{T}_t,(i,j)}^2,$$

(5)

where $\boldsymbol{E}_{\mathcal{T}_t} \in \mathbb{R}^{d_O \times d_I}$, $\Delta \boldsymbol{W}_{\mathcal{T}_t,(i,j)} = \boldsymbol{W}_{\mathcal{T}_t,(i,j)} - \boldsymbol{W}_{\mathcal{T}_t,\neg(i,j)}$. We define the first-order causal effect as the term involving only the first-order derivative, while the second-order causal effect includes both first- and second-order terms. Since directly computing the second-order term incurs quadratic complexity, we employ a diagonal Hessian approximation in practice [48], which reduces the computational cost to linear time [47]. This way enables scalable computation for both causal effect approximations. When $\boldsymbol{E}_{\mathcal{T}_t,(i,j)}$ is positive, it indicates that removing the parameter will lead to an increase in the task loss, suggesting that the parameter is causally essential for improving task performance. Conversely, when $\boldsymbol{E}_{\mathcal{T}_t,(i,j)}$ is negative, the parameter is ineffective. Finally, to map the causal effect $\boldsymbol{E}_{\mathcal{T}_t}$ to a weight between 0 and 1, we apply the softmax function to get $\widehat{\boldsymbol{E}}_{\mathcal{T}_t} = \text{Softmax}(\boldsymbol{E}_{\mathcal{T}_t})$. A larger value of $\widehat{\boldsymbol{E}}_{\mathcal{T}_t}(i,j)$ reflects a stronger causal effect of the parameter on task performance. Prioritizing such effective parameters during the current task training not only reduces the influence of ineffective ones but also enhances the model's ability to learn further tasks.

### 3.3 Cross-task Gradient Adaptation

After obtaining the causal effects of the parameters, another key challenge is accurately estimating the correlation between the new task and old tasks. This is crucial to ensure that the causal effects can be effectively integrated with task correlation to refine old tasks appropriately. To achieve this, we propose a **c**ross-t**a**sk **g**radient **a**daptation (CaGA) method that selectively updates parameters beneficial to old tasks when learning a new one. Specifically, we first quantify task correlation through cross-task gradient projection to evaluate the possibility of knowledge transfer. Then, we compute task affinity via gradient similarity to assess whether the transfer is beneficial (positive transfer) or detrimental (negative transfer). Based on these measures, we identify the gradient components of the new task that are most advantageous to old tasks. Finally, we perform a targeted update by emphasizing the gradient components that are causally effective for the new task and those that are beneficial to old tasks, thereby enabling potential backward knowledge transfer.

**Task Correlation.** To determine whether the new task meets the conditions for backward knowledge transfer to old tasks, we characterize the correlation between the input subspaces of old and new tasks using gradient projection [15, 16]. Specifically, given any new task $\mathcal{T}_t$ ($t \in [2, T]$) with gradient $\boldsymbol{G}_{\mathcal{T}_t} \in \mathbb{R}^{d_O \times d_I}$, its correlation with a old task $\mathcal{T}_k$ ($k \in [1, t-1]$) is defined as $C_{\mathcal{T}_t \to \mathcal{T}_k}$:

$$\boldsymbol{P}_{\mathcal{T}_t \to \mathcal{T}_k} = \boldsymbol{U}_{\mathcal{T}_k} \boldsymbol{U}_{\mathcal{T}_k}^\top \boldsymbol{G}_{\mathcal{T}_t}, \quad C_{\mathcal{T}_t \to \mathcal{T}_k} = \frac{\|\boldsymbol{P}_{\mathcal{T}_t \to \mathcal{T}_k}\|_2}{\|\boldsymbol{G}_{\mathcal{T}_t}\|_2}, \quad C_{\mathcal{T}_t \to \mathcal{T}_k} \in (0,1), \tag{6}$$

where $\boldsymbol{P}_{\mathcal{T}_t \to \mathcal{T}_k} \in \mathbb{R}^{d_O \times d_I}$ denotes the projection of the gradient of $\mathcal{T}_t$ onto the input subspace of $\mathcal{T}_k$, and $\boldsymbol{U}_{\mathcal{T}_k} = [\boldsymbol{u}_1, ..., \boldsymbol{u}_r] \in \mathbb{R}^{d_O \times r}$ is the base for the input subspace of $\mathcal{T}_k$ obtained via singular value decomposition (SVD) (see Appendix A for details). Intuitively, the norm of projected gradient (i.e., $\|\boldsymbol{P}_{\mathcal{T}_t \to \mathcal{T}_k}\|_2$) serves as a proxy for the correlation between the input subspaces of the two tasks, because the gradient lies within the span of the input features [6, 15, 16, 49]. A higher correlation score $C_{\mathcal{T}_t \to \mathcal{T}_k}$ indicates that $\boldsymbol{G}_{\mathcal{T}_t}$ projects strongly onto the input subspace of $\mathcal{T}_k$, suggesting substantial input subspace overlap and favorable conditions for backward knowledge transfer [15, 16].

**Task Affinity.** We introduce task affinity to assess whether the knowledge transfer from the new task to old tasks is beneficial. Since the similarity of task gradients within a shared parameter space can reflect inter-task relationships [46, 50], we quantify these relationships by computing the directional similarity between the projected gradient component of the new task onto the input space of the old tasks and the gradient of the old tasks themselves. Specifically, for any new task $\mathcal{T}_t(t \in [2, T])$, we define the task affinity matrix $\widehat{\boldsymbol{S}}_{\mathcal{T}_t \to \mathcal{T}_k} \in \mathbb{R}^{d_O \times d_I}$ to denote whether it has a positive transfer to the

old task $\mathcal{T}_k(k \in [1, t-1])$. The $j$-th column vector $\widehat{\boldsymbol{S}}_{\mathcal{T}_t \to \mathcal{T}_k, (:,j)}$ is defined as Eq. (7):

$$\boldsymbol{S}_{\mathcal{T}_t \to \mathcal{T}_k, (j)} = \frac{\boldsymbol{P}_{\mathcal{T}_t \to \mathcal{T}_k, (:,j)}}{\|\boldsymbol{P}_{\mathcal{T}_t \to \mathcal{T}_k, (:,j)}\|_2} \cdot \frac{\boldsymbol{G}_{\mathcal{T}_k, (:,j)}}{\|\boldsymbol{G}_{\mathcal{T}_k, (:,j)}\|_2}, \quad \widehat{\boldsymbol{S}}_{\mathcal{T}_t \to \mathcal{T}_k, (:,j)} = \begin{cases} +\mathbf{1}_{d_O} \in \mathbb{R}^{d_O}, & \text{if } \boldsymbol{S}_{\mathcal{T}_t \to \mathcal{T}_k, (j)} > 0 \\ \mathbf{0}_{d_O} \in \mathbb{R}^{d_O}, & \text{if } \boldsymbol{S}_{\mathcal{T}_t \to \mathcal{T}_k, (j)} = 0 \\ -\mathbf{1}_{d_O} \in \mathbb{R}^{d_O}, & \text{if } \boldsymbol{S}_{\mathcal{T}_t \to \mathcal{T}_k, (j)} < 0 \end{cases},$$
(7)

where $\boldsymbol{G}_{\mathcal{T}_k, (:,j)}$ denotes the $j$-th column vector of the gradient of $\mathcal{T}_k$ and $\boldsymbol{S}_{\mathcal{T}_t \to \mathcal{T}_k, (j)} \in [-1, 1]$ measures the similarity between the $j$-th column vectors of $\boldsymbol{P}_{\mathcal{T}_t \to \mathcal{T}_k}$ and $\boldsymbol{G}_{\mathcal{T}_k}$ in the input subpace of $\mathcal{T}_k$. The set $\{+\mathbf{1}_{d_O}, \mathbf{0}_{d_O}, -\mathbf{1}_{d_O}\}$ denotes all possible values of task affinity, where each is a $d_O$-dimensional column vectors with entries in $\{+1, 0, -1\}$, used to indicate the affinity between corresponding columns of the new and old task gradients.

**Adaptive Gradient Correction.** To enable effective backward knowledge transfer, we propose a adaptive gradient correction strategy to selectively update parameters under three cases: (1) If the new task exhibits no significant correlation to old tasks (indicating no potential for backward knowledge transfer) or the task affinity is zero, the strategy focuses on updating effective parameters with strong causal effects, without considering the impact on old tasks; (2) If the new task is strongly correlated to the old tasks (indicating potential for transfer), but the task affinity is negative, suggesting that the new task interferes with the old tasks, the strategy protects the parameters of the old tasks to prevent catastrophic forgetting; (3) If the new task demonstrates both strong correlation and positive task affinity with the old tasks (enabling effective backward knowledge transfer), it selectively updates the parameters that benefit the old tasks, thereby facilitating knowledge migration. To implement these adaptive updates, we introduce a gradient correction term, denoted as $\widehat{\boldsymbol{G}}_{\mathcal{T}_t}$:

$$\widehat{\boldsymbol{P}}_{\mathcal{T}_t \to \mathcal{T}_k} = C_{\mathcal{T}_t \to \mathcal{T}_k} \cdot \frac{\boldsymbol{P}_{\mathcal{T}_t \to \mathcal{T}_k}}{\|\boldsymbol{P}_{\mathcal{T}_t \to \mathcal{T}_k}\|_2}, \quad \widehat{\boldsymbol{G}}_{\mathcal{T}_t} = \widehat{\boldsymbol{E}}_{\mathcal{T}_t} \circ \boldsymbol{G}_{\mathcal{T}_t} \circ \left[ \mathbf{1}_{d_O \times d_I} + \sum_{k=1}^{t-1} \widehat{\boldsymbol{S}}_{\mathcal{T}_t \to \mathcal{T}_k} \circ \widehat{\boldsymbol{P}}_{\mathcal{T}_t \to \mathcal{T}_k} \right], \quad (8)$$

where $\widehat{\boldsymbol{P}}_{\mathcal{T}_t \to \mathcal{T}_k}$ denotes the gradient projection of $\mathcal{T}_t$, weighted by its correlation with the input space of $\mathcal{T}_k$, capturing the extent to which optimizing the parameter space of $\mathcal{T}_t$ affects that of $\mathcal{T}_k$. $\mathbf{1}_{d_O \times d_I}$ denotes an all-ones matrix of dimension $d_O \times d_I$, $\cdot$ indicates scalar multiplication, and $\circ$ represents the element-wise (Hadamard) product of matrices. Finally, the LoRA of task $\mathcal{T}_t$ updates its parameter matrices $\boldsymbol{A}_{\mathcal{T}_t}$ and $\boldsymbol{B}_{\mathcal{T}_t}$ at each training step $s$ using the corrected gradient defined in Eq. (9).

$$\boldsymbol{A}_{\mathcal{T}_t, (s+1)} \leftarrow \boldsymbol{A}_{\mathcal{T}_t, (s)} - \eta \boldsymbol{B}_{\mathcal{T}_t, (s)}^\top \widehat{\boldsymbol{G}}_{\mathcal{T}_t}, \quad \boldsymbol{B}_{\mathcal{T}_t, (s+1)} \leftarrow \boldsymbol{B}_{\mathcal{T}_t, (s)} - \eta \widehat{\boldsymbol{G}}_{\mathcal{T}_t} \boldsymbol{A}_{\mathcal{T}_t, (s)}^\top. \quad (9)$$

In the Eq. (8), the corrected gradient $\widehat{\boldsymbol{G}}_{\mathcal{T}_t}$ can adaptively adjust the gradient of task $\mathcal{T}_t$ at each iteration to achieve better knowledge transfer. Let $\boldsymbol{G}_{\mathcal{T}_t, (:,j)}$ denote the $j$-th column vector of the gradient for task $\mathcal{T}_t$. When the values of $\widehat{\boldsymbol{P}}_{\mathcal{T}_t \to \mathcal{T}_k, (:,j)}$ are very small (i.e., $\widehat{\boldsymbol{G}}_{\mathcal{T}_t, (:,j)} \approx \widehat{\boldsymbol{E}}_{\mathcal{T}_t, (:,j)} \circ \boldsymbol{G}_{\mathcal{T}_t, (:,j)}$) or when $\widehat{\boldsymbol{S}}_{\mathcal{T}_t \to \mathcal{T}_k, (:,j)}$ is a zero vector (i.e., $\widehat{\boldsymbol{G}}_{\mathcal{T}_t, (:,j)} = \widehat{\boldsymbol{E}}_{\mathcal{T}_t, (:,j)} \circ \boldsymbol{G}_{\mathcal{T}_t, (:,j)}$), it indicates that the new task has no significant influence on old tasks. In this case, parameter updates based on $\widehat{\boldsymbol{G}}_{\mathcal{T}_t, (:,j)}$ can proceed only based on causal effects, without interfering with old task knowledge. In contrast, when $\widehat{\boldsymbol{P}}_{\mathcal{T}_t \to \mathcal{T}_k, (:,j)}$ contains large values and $\widehat{\boldsymbol{S}}_{\mathcal{T}_t \to \mathcal{T}_k, (:,j)}$ is a vector of -1, it indicates that a conflict between new ans old tasks. To resolve this, the corrected gradient is computed as $\widehat{\boldsymbol{G}}_{\mathcal{T}_t, (:,j)} = \widehat{\boldsymbol{E}}_{\mathcal{T}_t, (:,j)} \circ \left[ \boldsymbol{G}_{\mathcal{T}_t, (:,j)} - \sum_{k=1}^{t-1} \widehat{\boldsymbol{P}}_{\mathcal{T}_t \to \mathcal{T}_k, (:,j)} \circ \boldsymbol{G}_{\mathcal{T}_t, (:,j)} \right]$, which removes gradient components aligned with old tasks. Updating the parameters based on this residual gradient helps mitigate catastrophic forgetting. Finally, when the values in $\widehat{\boldsymbol{P}}_{\mathcal{T}_t \to \mathcal{T}_k, (:,j)}$ are large, and $\widehat{\boldsymbol{S}}_{\mathcal{T}_t \to \mathcal{T}_k, (:,j)}$ is a vector of 1, it indicates that the new task positively contributes to old tasks. In this case, $\widehat{\boldsymbol{G}}_{\mathcal{T}_t, (:,j)}$ selectively incorporates advantageous components from $\boldsymbol{G}_{\mathcal{T}_t, (:,j)}$ to promote backward knowledge transfer.

## 3.4 Algorithm and Time Complexity

**Algorithm.** CaLoRA adaptively updates the LoRA parameters based on the gradient correction terms during the training process. As shown in Algorithm 1, given the data $\mathcal{D}_t$ of the current task $t$, CaLoRA first estimates the causal effect of the LoRA parameters at the current step $s$. For new tasks $\mathcal{T}_t(t > 1)$, it calculates the correlation and affinity between the new task and the old tasks at the

---
**Algorithm 1** CaLoRA for Continual Learning
---
**Input:** datasets: $\mathcal{D} = \{\mathcal{D}_t\}_{t=1}^{t=T}$, pretrained model $f_{\boldsymbol{W}}(\cdot)$, training steps S
**Output:** model $f_{\boldsymbol{W}}(\cdot)$ with learned parameters $\{\boldsymbol{W}_{\mathcal{T}_t}\}_{t=1}^T$
1: **for** $\mathcal{D}_t$ in $\mathcal{D}$ **do**
2:     Initialize $\boldsymbol{A}_{\mathcal{T}_t}$ and $\boldsymbol{B}_{\mathcal{T}_t}$.
3:     **for** $s \in [1, \ldots, S]$ **do**
4:         Estimate causal effect $\widehat{\boldsymbol{E}}_{\mathcal{T}_t,(s)}$ at step $s$ by $\widehat{\boldsymbol{E}}_{\mathcal{T}_t,(s)} = \text{Softmax}(\boldsymbol{E}_{\mathcal{T}_t,(s)})$.
5:         **if** $t > 1$ **then**
6:             Measure task correlation $C_{\mathcal{T}_t \rightarrow \mathcal{T}_k,(s)}$ at step $s$ by Eq. (6).
7:             Quantify task affinity $\widehat{S}_{\mathcal{T}_t \rightarrow \mathcal{T}_k,(s)}$ at step $s$ by Eq. (7).
8:             Calculate gradient correction term $\widehat{\boldsymbol{G}}_{\mathcal{T}_t,(s)}$ at step $s$ by Eq. (8).
9:         **end if**
10:       Update $\boldsymbol{A}_{\mathcal{T}_t,(s+1)}, \boldsymbol{B}_{\mathcal{T}_t,(s+1)}$ using Eq. (9).
11:       Save the optimal task gradient in the task memory.
12:     **end for**
13: **end for**
---

current step $s$. Finally, CaLoRA adaptively updates the LoRA parameters based on the causal effect, task correlation, and affinity.

**Time Complexity Analysis.** Given a pretrained weight matrix $W \in \mathbb{R}^{d_O \times d_I}$, PaCA estimates causal effects via a second-order Taylor expansion with a diagonal Hessian approximation. Both the first-order and second-order terms have a time complexity of $\mathcal{O}(d_O \times d_I)$, making the overall complexity of PaCA linear and comparable to a standard backward pass. This computation can be efficiently parallelized and seamlessly integrated into the training process. In a task sequence of $T$ tasks, for each new task, each gradient projection and affinity computation costs $\mathcal{O}(d_O \times d_I)$, leading to a total overhead of $\mathcal{O}((t-1) \times d_O \times d_I)$. Thus, the per-task complexity is approximately $\mathcal{O}(t \times d_O \times d_I)$. The total complexity for the entire sequence is $\mathcal{O}\left(\sum_{t=1}^T t \times d_O \times d_I\right)$.

## 4 Experiment

### 4.1 Experimental Setups

**Dataset.** We conduct experiments on two natural language processing (NLP) benchmarks (i.e, SuperNI [51] and Long Sequence [52]) and a computer vision (CV) benchmark (i.e., ImageNet-R [53]). Specifically, following prior work [13], both the SuperNI and Long Sequence benchmarks consist of 15 tasks, each evaluated under two different task orders. ImageNet-R is derived from 200 ImageNet classes through various artistic transformations. For the ImageNet-R benchmark, in line with previous studies [6, 7, 20], it is split into 5, 10, and 20 tasks, corresponding to 40, 20, and 10 classes per task, respectively. Further details on the benchmarks are provided in Appendix B.

**Metrics.** Let $Acc_{\mathcal{T}_t, \mathcal{T}_k}$ denote the testing performance (Accuracy for classification tasks and Rouge-L [54] for other tasks, more details are provided in the Appendix B) on the $k$-th task after training $t$-th task. The evaluation metrics are defined as follows: (1) Average Performance (AP) [55] reflects the mean performance across all tasks after training on the final task, i.e., $AP = \frac{1}{T} \sum_{t=1}^T Acc_{\mathcal{T}_T, \mathcal{T}_t}$; (2) Forgetting Rate (F.Ra) [55] denotes the extent of knowledge forgotten across the first $t-1$ tasks, i.e., $F.Ra = \frac{1}{T-1} \sum_{t=1}^{T-1} \left(\max_{k=1}^{T-1} Acc_{\mathcal{T}_k, \mathcal{T}_t} - Acc_{\mathcal{T}_T, \mathcal{T}_t}\right)$; (3) Forward Transfer (FWT) [56] measures the influence of old task knowledge on learning new tasks, i.e., $FWT = \frac{1}{T} \sum_{t=1}^T Acc_{\mathcal{T}_t, \mathcal{T}_t} - Acc_{\mathcal{T}_t}$, where $Acc_{\mathcal{T}_t}$ refers to the performance of training task $\mathcal{T}_t$ individually; (4) Backward Transfer (BWT) [57] assesses the impact of learning new tasks on old tasks, i.e., $BWT = \frac{1}{T-1} \sum_{t=1}^{T-1} \left(Acc_{\mathcal{T}_T, \mathcal{T}_t} - Acc_{\mathcal{T}_t, \mathcal{T}_t}\right)$.

**Comparison Baselines and Training Details.** We evaluate CaLoRA against nine PEFT-based continual learning baselines, including SeqLoRA, L2P [1], CodaPrompt [20], HidePrompt [21], O-LoRA [12], InfLoRA [6], SAPT-P [13], SAPT-LoRA [13], and SD-LoRA [7]. CaLoRA is a

Table 2: Overall results (mean ± std over 3 random seeds) on SuperNI benchmark with two task orders, evaluated using the T5-Large model. **Bold** indicates the best values, underline presents the second-best values.

| Method | Order 1 with 15 tasks | | | | Order 2 with 15 tasks | | | |
|---|---|---|---|---|---|---|---|---|
| | AP±std↑ | F.Ra±std↓ | FWT±std↑ | BWT±std↑ | AP±std↑ | F.Ra±std↓ | FWT±std↑ | BWT±std↑ |
| SeqLoRA | 5.41±0.54 | 30.73±0.71 | -17.32±0.18 | -28.54±0.86 | 7.86±0.62 | 35.55±0.51 | -9.76±0.16 | -30.81±0.78 |
| L2P | 15.65±0.45 | 7.12±0.41 | -19.86±0.89 | -3.33±0.23 | 9.97±0.31 | 17.50±0.66 | -16.94±0.74 | -12.20±0.32 |
| CodaPrompt | 21.31±0.24 | 5.12±0.31 | -0.86±0.15 | -3.31±0.44 | 16.11±0.74 | 12.24±0.61 | -7.65±0.93 | -10.37±0.54 |
| HidePrompt | 26.45±0.34 | 3.61±0.25 | -0.17±0.14 | -3.42±0.37 | 25.22±0.81 | 10.33±0.33 | -3.21±0.52 | -5.01±0.23 |
| O-LoRA | 23.66±0.93 | 29.22±0.72 | -0.41±0.31 | -24.38±1.03 | 27.21±0.88 | 21.25±0.96 | 0.44±0.23 | -18.96±0.73 |
| InfLoRA | 43.56±0.63 | 1.45±0.45 | 0.15±0.32 | -2.51±0.26 | 41.82±0.55 | 1.88±0.13 | 1.14±0.24 | -2.71±0.36 |
| SAPT-P | 42.38±0.56 | 1.72±0.34 | 2.52±0.44 | -1.11±0.27 | 41.55±0.68 | 1.77±0.36 | 1.01±0.35 | -0.94±0.26 |
| SAPT-LoRA | 51.76±0.71 | 0.88±0.11 | 2.21±0.41 | -0.75±0.25 | 50.09±0.36 | 1.76±0.23 | 1.54±0.33 | -1.28±0.41 |
| SD-LoRA | 45.91±0.53 | 1.31±0.26 | 2.11±0.31 | -2.13±0.22 | 44.37±0.76 | 1.71±0.19 | 1.55±0.36 | -2.06±0.31 |
| CaLoRA | **54.42**±0.55 | **0.25**±0.14 | **2.98**±0.21 | **0.35**±0.21 | **52.76**±0.67 | **0.85**±0.11 | **2.22**±0.33 | **0.18**±0.19 |

model-agnostic continual learning framework compatible with any transformer-based pre-trained model (e.g., T5, LLaMA-2, and ViT). Following prior works [13, 7], CaLoRA is applied to the query and value projections within the attention modules of all Transformer blocks in each pretrained model. To ensure a fair comparison with recent works on the two NLP benchmarks, we implement CaLoRA based on the pre-trained T5 (0.77B and 3B) [58] and LLaMA-2 (7B) [59] models. The ranks of LoRA are set to 4 and 8 for the LLaMA-2 and T5 models, respectively. For the ImageNet-R benchmark, following existing works [7], we use the ViT-B/16 backbone [60] supervised pre-trained on ImageNet 21K as the pre-trained model, and the rank of LoRA is set to 10. All experiments are conducted with 3 H800 and 4 V100 GPUs. More training details are provided in Appendix C.

## 4.2 Main Results

**Results on Different Task Orders and Model Scales.** Tables 2 and 3 report continual learning performance under different task orders on two NLP benchmarks. CaLoRA consistently outperforms state-of-the-art baselines, including SAPT-P, SAPT-LoRA, and SD-LoRA. Specifically, Table 2 highlights CaLoRA's superior performance on the SuperNI benchmark. Under Order 1, it achieves an average performance gain of 2.66, with improvements of 0.46 in forward transfer and 1.1 in backward transfer, while reducing forgetting rate by 0.63. Similar gains are observed under Order 2, with increases of 2.67, 0.67, and 1.12 in average performance, forward transfer, and backward transfer, respectively, and a 0.86 reduction in forgetting rate. Table 3 shows that CaLoRA also excels on the Long Sequence benchmark. For Order 1, it improves average performance by 1.81, forward transfer by 0.83, and backward transfer by 1.44, while reducing forgetting rate by 1.05. Under Order 2, it achieves gains of 1.29, 0.88, and 1.34 in the respective metrics, with a 0.91 reduction in forgetting rate. Figure 2 further compares CaLoRA against three competitive baselines across three scaled

Table 3: Overall results (mean ± std over 3 random seeds) on Long Sequence benchmark with two task orders, evaluated using the T5-Large model. **Bold** indicates the best values, underline presents the second-best values.

| Method | Order 1 with 15 tasks | | | | Order 2 with 15 tasks | | | |
|---|---|---|---|---|---|---|---|---|
| | AP±std↑ | F.Ra±std↓ | FWT±std↑ | BWT±std↑ | AP±std↑ | F.Ra±std↓ | FWT±std↑ | BWT±std↑ |
| SeqLoRA | 7.34±0.45 | 80.44±1.13 | 0.99±0.24 | -76.10±0.97 | 14.11±0.55 | 74.88±0.79 | 0.77±0.21 | -71.24±0.89 |
| L2P | 58.35±0.71 | 20.31±0.65 | 1.23±0.21 | -15.1±0.34 | 58.10±0.61 | 22.76±0.42 | 1.55±0.31 | -18.34±0.56 |
| CodaPrompt | 65.33±0.77 | 11.21±0.35 | 1.87±0.26 | -7.65±0.46 | 64.43±0.89 | 13.35±0.56 | 1.67±0.31 | -10.31±0.67 |
| HidePrompt | 69.78±0.41 | 7.85±0.25 | 2.11±0.14 | -4.01±0.35 | 69.12±0.86 | 6.43±0.46 | 1.81±0.32 | -7.21±0.44 |
| O-LoRA | 70.14±0.64 | 7.87±0.24 | -7.10±0.32 | -4.04±0.31 | 70.26±0.55 | 5.11±0.30 | -7.88±0.21 | -5.56±0.37 |
| InfLoRA | 80.26±0.76 | 2.21±0.13 | 0.65±0.27 | -3.27±0.45 | 78.21±0.64 | 2.99±0.31 | 0.38±0.32 | -4.05±0.24 |
| SAPT-P | 79.84±0.45 | 2.45±0.14 | 3.35±0.24 | -1.45±0.21 | 77.68±0.61 | 2.76±0.24 | 2.26±0.29 | -2.04±0.26 |
| SAPT-LoRA | 82.81±0.66 | 1.21±0.13 | 2.23±0.21 | -1.15±0.16 | 80.66±0.72 | 2.46±0.33 | 2.01±0.32 | -1.91±0.25 |
| SD-LoRA | 81.35±0.55 | 1.76±0.32 | 2.51±0.34 | -2.71±0.29 | 78.98±0.68 | 3.21±0.33 | 2.33±0.39 | -2.77±0.31 |
| CaLoRA | **84.62**±0.63 | **0.16**±0.55 | **4.18**±0.39 | **0.29**±0.13 | **81.95**±0.49 | **1.55**±0.32 | **3.21**±0.45 | **-0.57**±0.12 |

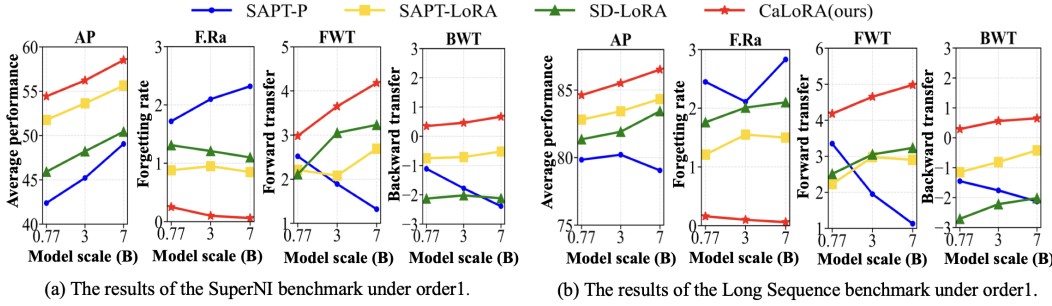

(a) The results of the SuperNI benchmark under order1.

(b) The results of the Long Sequence benchmark under order1.

Figure 2: Comparative performance of CaLoRA against suboptimal baselines across three scaled pretrained models: T5-Large (0.77B), T5-XL (3B), and LLaMA-2 (7B).

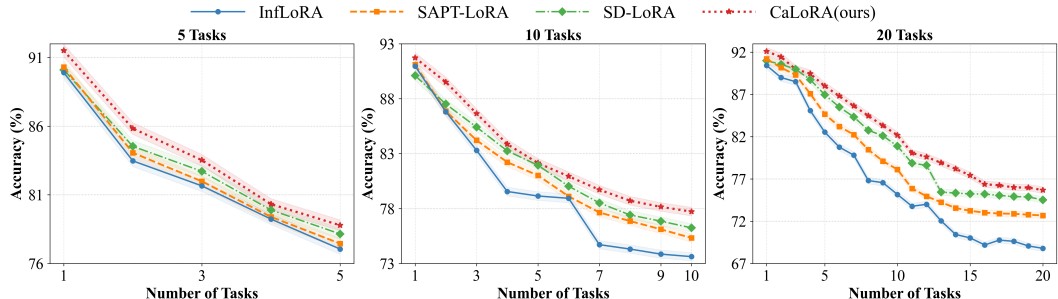

Figure 3: Variation of the performance of different methods during the learning of ImageNet-R.

pre-trained models on two NLP benchmarks in the order 1. The results demonstrate that CaLoRA consistently delivers superior continual learning performance regardless of model size.

**Results across Varied Task Lengths.** Table 4 presents the comparative results on the ImageNet-R benchmark for both 10-task and 20-task continual learning scenarios. CaLoRA demonstrates consistent superiority over all baseline methods in both task lengths. In the 10-task setting, CaLoRA surpasses the second-best baselines (i.e., SD-LoRA and SAPT-LoRA) with improvements: 1.48 in average performance, 0.67 in forward transfer, and 1.4 in backward transfer, while simultaneously achieving a 0.78 reduction in forgetting rate. The advantages persist in the more challenging 20-task setting, where CaLoRA maintains leads of 1.16, 0.73, and 0.97 in the respective metrics, along with a 0.57 decrease in forgetting rate. Complementing these quantitative results, Figure 3 visually demonstrates CaLoRA's superior learning trajectory compared to three suboptimal continual learning methods on the ImageNet-R benchmark. The accuracy curves reveal that CaLoRA maintains consistently higher performance throughout the entire learning process, not just at the final evaluation stage. Additional supporting results are provided in Appendix D. In particular, these results demonstrate that CaLoRA more effectively mitigates catastrophic forgetting by enabling positive backward knowledge transfer, in comparison to several baseline methods.

Table 4: Overall results (mean ± std over 3 random seeds) on ImageNet-R (10/20 tasks) benchmark with the ViT-B/16 backbone. **Bold** (underline) indicates the best (second-best) values.

| Method | 10 tasks | | | | 20 tasks | | | |
|---|---|---|---|---|---|---|---|---|
| | AP±std↑ | F.Ra±std↓ | FWT±std↑ | BWT±std↑ | AP±std↑ | F.Ra±std↓ | FWT±std↑ | BWT±std↑ |
| SeqLoRA | 62.45±0.68 | 22.41±0.58 | 0.86±0.21 | -17.61±0.33 | 50.86±0.43 | 32.16±0.68 | 0.73±0.13 | -14.22±0.37 |
| L2P | 67.66±0.48 | 15.15±0.27 | 1.07±0.21 | -13.69±0.71 | 64.64±0.41 | 19.81±0.63 | 0.79±0.19 | -12.75±0.63 |
| CodaPrompt | 71.28±0.41 | 9.67±0.32 | 1.71±0.23 | -9.32±0.47 | 67.25±0.33 | 11.83±0.45 | 1.21±0.17 | -8.46±0.54 |
| HidePrompt | 72.62±0.34 | 8.38±0.45 | 1.76±0.23 | -10.33±0.57 | 69.88±0.39 | 9.36±0.53 | 1.43±0.31 | -8.73±0.42 |
| O-LoRA | 72.15±0.45 | 5.32±0.36 | -0.97±0.24 | -6.42±0.31 | 66.23±0.28 | 7.81±0.63 | -0.81±0.17 | -5.81±0.37 |
| InfLoRA | 73.62±0.34 | 4.11±0.35 | 0.56±0.23 | -4.16±0.43 | 68.79±0.36 | 6.89±0.35 | 0.67±0.21 | -4.39±0.34 |
| SAPT-P | 72.82±0.54 | 5.33±0.41 | 2.26±0.45 | -2.21±0.37 | 68.08±0.35 | 7.95±0.45 | 2.15±0.31 | -1.16±0.31 |
| SAPT-LoRA | 75.32±0.54 | 3.25±0.35 | 3.23±0.41 | -1.25±0.32 | 72.68±0.46 | 5.25±0.33 | 3.21±0.21 | -0.89±0.13 |
| SD-LoRA | 76.24±0.32 | 3.11±0.31 | 3.51±0.32 | -1.71±0.23 | 74.52±0.43 | 5.31±0.35 | 3.25±0.26 | -1.94±0.23 |
| CaLoRA | **77.72**±0.36 | **2.33**±0.27 | **4.18**±0.33 | **0.15**±0.19 | **75.68**±0.45 | **4.68**±0.31 | **3.98**±0.23 | **0.08**±0.21 |

Table 5: Ablation results on SuperNI benchmark with two task orders, evaluated using the T5-Large model. **Bold** (underline) indicates the best (second-best) values.

| Parameter Updating Strategy | Order 1 with 15 tasks | | | | Order 2 with 15 tasks | | | |
|---|---|---|---|---|---|---|---|---|
| | AP ↑ | F.Ra ↓ | FWT ↑ | BWT ↑ | AP↑ | F.Ra ↓ | FWT ↑ | BWT ↑ |
| w/o Task Correlation (TaC) | 53.15 | 0.61 | 1.97 | -0.42 | 50.83 | 1.33 | 2.01 | -0.61 |
| w/o Task Affinity (TaA) | 51.41 | 0.95 | 1.52 | -1.19 | 47.66 | 2.83 | 1.53 | -1.26 |
| w/ Causal Effect (CaE) | 44.86 | 2.18 | 0.41 | -3.45 | 43.72 | 2.83 | 0.51 | -2.21 |
| w/ CaE+Gradient Projection | 45.05 | 1.81 | 0.93 | -1.95 | 44.13 | 1.73 | 0.76 | -1.77 |
| w/ Gradient Projection (GradProj) | 43.35 | 1.86 | 0.53 | -2.12 | 42.16 | 2.33 | 0.21 | -2.28 |
| w/ GradProj+TaC | 45.57 | 1.61 | 1.21 | -1.23 | 44.36 | 1.71 | 1.06 | -1.66 |
| w/ GradProj+TaC+TaA (w/o CaE) | 48.89 | 1.08 | 2.23 | -0.54 | 47.21 | 1.66 | 2.04 | -0.71 |
| CaLoRA | **54.42** | **0.25** | **2.98** | **0.35** | **52.76** | **0.85** | **2.22** | **0.18** |

## 4.3 Ablation Studies.

To evaluate the impact of each design component in CaLoRA, we conduct ablation studies using seven parameter update strategies, with the results summarized in Table 5. The ablation settings are as follows: (1) w/o Task Correlation (TaC) and (2) w/o Task Affinity (TaA) remove task correlation and affinity when optimizing the gradient, respectively; (3) w/ Causal Effect only uses the causal effect to constrain the task gradient; (4) w/ CaE+Gradient projection applies both causal effect and naïve gradient projection, without task correlation or affinity; (5) w/ Gradient Projection (GradProj) employs only naïve gradient projection, excluding causal effect, task correlation, and affinity; (6) w/ GradProj+TaC combines gradient projection and task correlation, without task affinity or causal effect; (7) w/ GradProj+TaC+TaA incorporates gradient projection, task correlation, and affinity, excluding causal effect. As shown in Table 5, the causal effect significantly improves average performance, while gradient projection, task correlation, and task affinity are essential for alleviating forgetting and enhancing both forward and backward knowledge transfer.

## 5 Conclusion

In this work, we go beyond the conventional focus on mitigating catastrophic forgetting and explore backward knowledge transfer in PEFT-based continual learning. To this end, we propose CaLoRA, a novel **c**ausal-**a**ware **lo**w-**r**ank **a**daptation framework explicitly designed to facilitate backward knowledge transfer. We introduce **pa**rameter-level **c**ounterfactual **a**ttribution (PaCA) to estimate the causal effect of parameters, and **c**ross-**ta**sk **g**radient **a**daptation (CaGA) to estimate task correlation and affinity. CaGA adaptively adjusts gradients based on these measures, enabling backward knowledge transfer without data replay. Extensive experiments on multiple benchmarks demonstrate that CaLoRA consistently outperforms existing state-of-the-art methods.

**Limitations.** CaLoRA has two limitations: (1) Although it avoids data replay, it requires storing old task gradients, which can lead to increased memory overhead when the number of tasks or parameter dimensionality is large. Future work will explore task correlation based on parameter function space modeling; (2) The method assumes clear task boundaries, which limits its applicability. Future work will extend CaLoRA to more general online continual learning scenarios by modeling perturbations in the space of task parameters.

**Broader Impacts.** In this work, we propose CaLoRA to explore backward knowledge transfer in PEFT-based continual learning. We believe that this work will have a positive social impact. By improving the efficiency and scalability of continual learning, CaLoRA has the potential to reduce the computational cost and energy consumption associated with retraining large models from scratch, thus contributing to more sustainable AI development.

## 6 Acknowledgment

This work was supported in part by the National Key Research and Development Program of China (Grant No. 2023YFB3107000) and the Major Key Project of PCL (Grant No. PCL2024A05).

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

# A Singular Value Decomposition

Let $\boldsymbol{G} \in \mathbb{R}^{d_O \times d_I}$ denote a matrix (in our case, the gradient matrix of a task). The singular value decomposition of $\boldsymbol{G}$ is:

$$\boldsymbol{G} = \boldsymbol{U} \boldsymbol{\Sigma} \boldsymbol{V}^{\top}, \tag{10}$$

where $\boldsymbol{U} \in \mathbb{R}^{d_O \times d_O}$ contains the left singular vectors (column-orthonormal), $\boldsymbol{V} \in \mathbb{R}^{d_I \times d_I}$ contains the right singular vectors, $\boldsymbol{\Sigma} \in \mathbb{R}^{d_O \times d_I}$ is a diagonal matrix whose diagonal entries are the singular values $\{\sigma_1, \sigma_2, \ldots, \sigma_r\}$, with $\sigma_1 \geq \sigma_2 \geq \cdots \geq \sigma_r > 0$, and $r = \mathrm{rank}(\boldsymbol{G})$.

For a given task $\mathcal{T}_k$, let $\boldsymbol{G}_{\mathcal{T}_k} \in \mathbb{R}^{d_O \times d_I}$ denote the task gradient matrix. Applying singular value decomposition (SVD) to $\boldsymbol{G}_{\mathcal{T}_k}$, we obtain:

$$\boldsymbol{G}_{\mathcal{T}_k} = \boldsymbol{U}_{\mathcal{T}_k} \boldsymbol{\Sigma}_{\mathcal{T}_k} \boldsymbol{V}_{\mathcal{T}_k}^{\top}. \tag{11}$$

We then extract the top-$r$ left singular vectors $\boldsymbol{U}_{\mathcal{T}_k} = [\boldsymbol{u}_1, \boldsymbol{u}_2, \ldots, \boldsymbol{u}_r] \in \mathbb{R}^{d_O \times r}$, corresponding to the $r$ largest singular values, which capture the dominant directions of the gradient space. These vectors form an orthonormal basis that spans the input subspace relevant to task $\mathcal{T}_k$.

This gradient-based subspace projection method provides a theoretically grounded and computationally tractable method for quantifying task correlation [6, 15, 16]. By leveraging SVD, we extract meaningful input subspaces. By measuring the normalized projection of new gradients onto old subspaces, we obtain a reasonable task correlation that facilitates the analysis of backward knowledge transfer among tasks.

# B Additional Dataset Details

**Two NLP Benchmarks.** Following previous work [13], we adopt SuperNI and Long Sequence as the NLP benchmarks to evaluate continual learning methods for large language models (LLMs). These benchmarks' detailed descriptions and evaluation metrics are presented in Table 6. We consider two different task orders for each benchmark, as shown in Table 7. Specifically, the SuperNI Benchmark [51] comprises a diverse set of NLP tasks, each accompanied by expert-written instructions, facilitating rigorous and realistic evaluation in continual learning settings. It consists of 15 sequential tasks. For each task, 1,000 instances are randomly sampled for training, and 100 instances are used for validation and testing. The Long Sequence Benchmark [52] also includes 15 classification tasks, specifically designed for continual learning with LLMs. For each task, 1,000 training samples are randomly selected, and 500 samples per class are used for both validation and testing. This benchmark emphasizes challenges in handling long-context dependencies and task diversity in a sequential learning setting.

Table 6: The details of two NLP benchmarks.

| SuperNI Benchmark | | | Long Sequence Benchmark | | |
|---|---|---|---|---|---|
| Dataset name | Task | Metric | Dataset name | Task | Metric |
| 1. task639 | dialogue generation | Rouge-L | 1. Yelp | sentiment analysis | accuracy |
| 2. task1590 | dialogue generation | Rouge-L | 2. Amazon | sentiment analysis | accuracy |
| 3. task1729 | dialogue generation | Rouge-L | 3. DBpedia | topic classification | accuracy |
| 4. task181 | information extraction | Rouge-L | 4. Yahoo | topic classification | accuracy |
| 5. task748 | information extraction | Rouge-L | 5. AG News | topic classification | accuracy |
| 6. task1510 | information extraction | Rouge-L | 6. MNLI | natural language inference | accuracy |
| 7. task002 | question answering | Rouge-L | 7. QQP | paragraph detection | accuracy |
| 8. task073 | question answering | Rouge-L | 8. RTE | natural language inference | accuracy |
| 9. task591 | question answering | Rouge-L | 9. SST-2 | sentiment analysis | accuracy |
| 10. task511 | summarization | Rouge-L | 10. WiC | word sense disambiguation | accuracy |
| 11. task1290 | summarization | Rouge-L | 11. CB | natural language inference | accuracy |
| 12. task1572 | summarization | Rouge-L | 12. COPA | question and answering | accuracy |
| 13. task363 | sentiment analysis | accuracy | 13. BoolQA | boolean question and answering | accuracy |
| 14. task875 | sentiment analysis | accuracy | 14. MultiRC | question and answering | accuracy |
| 15. task1687 | sentiment analysis | accuracy | 15. IMDB | sentiment analysis | accuracy |

Table 7: Different task orders of two NLP benchmarks.

| Order | SuperNI Benchmark | Long Sequence Benchmark |
|---|---|---|
| 1 | task1572 → task363 → task1290 → task181 → task002 → task1510 → task639 → task1729 → task073 → task1590 → task748 → task511 → task591 → task1687 → task875 | MNLI → CB → WiC → COPA → QQP → BoolQA → RTE → IMDB → Yelp → Amazon → SST-2 → DBpedia → AG News → MultiRC → Yahoo |
| 2 | task748 → task073 → task1590 → task639 → task1572 → task1687 → task591 → task363 → task1510 → task1729 → task181 → task511 → task002 → task1290 → task875 | Yelp → Amazon → MNLI → CB → COPA → QQP → RTE → IMDB → SST-2 → DBpedia → AG News → Yahoo → MultiRC → BoolQA → WiC |

**The CV Benchmark.** We adopt the ImageNet-R dataset [53], which consists of 200 ImageNet [61] classes rendered through various artistic styles. Introduced into the continual learning by prior works [6, 7], ImageNet-R has become a widely used benchmark for evaluating parameter-efficient fine-tuning (PEFT) methods. Following prior works [6, 7, 20], we partition the dataset into 5, 10, and 20 tasks, corresponding to 40, 20, and 10 classes per task, respectively. These configurations allow for a comprehensive assessment of the scalability and adaptability of continual learning methods across varying levels of task granularity.

## C  Additional Baselines and Training Details

**Baselines.** We evaluate our proposed CaLoRA against nine PEFT-based continual learning baselines: (1) SeqLoRA sequentially trains LoRA modules following the task order; (2) L2P [1] dynamically selects and updates prompts from a fixed prompt pool based on the input; (3) CodaPrompt [20] composes prompts dynamically via attention mechanisms to mitigate forgetting and improve adaptability; (4) HidePrompt [21] introduces a hierarchical decomposition framework to enhance prompt-based continual learning; (5) O-LoRA [12] learns each task in an orthogonal LoRA subspace and aggregates LoRA weights at inference time; (6) InfLoRA [6] constructs task-specific LoRA subspaces to reduce interference between previously learned and new tasks; (7) SAPT-P (SAPT-Prompt) and (8) SAPT-LoRA are two variants of SAPT [13], which employ a shared attentive learning and selection module to align and select appropriate PEFT strategies based on prompt tuning and LoRA, respectively; (9) SD-LoRA [7] incrementally decouples the learning of direction and magnitude in LoRA parameters.

**Training Details.** All experiments are implemented using PyTorch [62] and the Transformers library [63]. Specifically, for the two NLP benchmarks, we use the AdamW [64] optimizer in T5 and LLaMA-2 with learning rates of 0.0003 and 0.0005, respectively. The experiments are conducted on three H800 GPUs to enhance computational efficiency, with per-GPU batch sizes set to 2 for LLaMA-2 and 16 for T5. For the SuperNI benchmark, the number of training epochs is set to 100 for T5 and 50 for LLaMA-2. For the LongSequence benchmark, the number of training epochs is set to 10 for T5 and 20 for LLaMA-2. For the ImageNet-R benchmark, we follow prior work [6] and use the Adam optimizer [65] with a learning rate of 0.0005. Each experiment is conducted on a single H800 or V100 GPU, with the batch size uniformly set to 128. Each task is trained for 50 epochs.

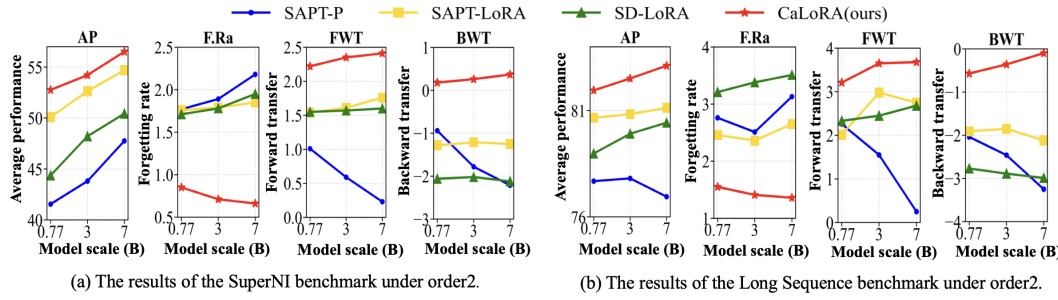

Figure 4: Comparative performance of CaLoRA against suboptimal baselines across three scaled pretrained models (T5-Large with 0.77B parameters, T5-XL with 3B parameters, and LLaMA-2 with 7B parameters) on two NLP benchmarks under order 2.

Table 8: Overall results (mean ± std over 3 random seeds) on ImageNet-R (5 tasks) with the ViT-B/16 backbone. **Bold** (underline) indicates the best (second-best) values.

| Method | 5 tasks | | | |
|---|---|---|---|---|
| | AP±std↑ | F.Ra±std↓ | FWT±std↑ | BWT±std↑ |
| SeqLoRA | 63.15±0.51 | 21.05±0.43 | 1.34±0.21 | -15.11±0.24 |
| L2P | 68.99±0.39 | 13.44±0.31 | 1.86±0.25 | -11.43±0.56 |
| CodaPrompt | 74.88±0.45 | 8.76±0.24 | 2.43±0.23 | -8.02±0.36 |
| HidePrompt | 74.46±0.41 | 8.01±0.33 | 2.76±0.26 | -8.33±0.31 |
| O-LoRA | 74.75±0.32 | 5.21±0.29 | -0.16±0.21 | -5.11±0.28 |
| InfLoRA | 75.82±0.41 | 3.78±0.35 | 0.98±0.34 | -3.88±0.35 |
| SAPT-P | 75.01±0.52 | 3.41±0.34 | 3.41±0.21 | -1.76±0.28 |
| SAPT-LoRA | 77.13±0.34 | 2.82±0.31 | 4.56±0.41 | -0.82±0.19 |
| SD-LoRA | 78.05±0.46 | 2.42±0.33 | 4.45±0.41 | -1.25±0.16 |
| CaLoRA | **79.11**±0.39 | **1.53**±0.25 | **4.98**±0.34 | **0.22**±0.19 |

Table 9: Comparison on ImageNet-R (20 tasks) in terms of computation (GFLOPs), parameters, and storage efficiency. † indicates that the results are from SD-LoRA [7].

| Method | GFLOPs | Learnable Parameters (M) | Stored Features (M) |
|---|---|---|---|
| L2P† | 70.14 | 0.48 | 0 |
| CodaPrompt† | 70.61 | 0.38 | 0 |
| HidePrompt† | 70.36 | 0.08 | 0.15 |
| O-LoRA | 35.12 | 0.19 | 0 |
| InfLoRA† | 35.12 | 0.37 | 0.10 |
| SAPT-P | 77.11 | 0.34 | 0.15 |
| SAPT-LoRA | 35.12 | 0.37 | 0.15 |
| SD-LoRA† | 35.12 | 0.37 | 0 |
| CaLoRA | 35.12 | 0.37 | 0.10 |

# D   Additional Results

**Additional Results of NLP and CV Benchmarks.** Figure 4 presents a comparison between CaLoRA and three strong baselines across three different scales of pre-trained models on two NLP benchmarks under task order 2. The results show that CaLoRA consistently achieves superior continual learning performance across all model sizes. Table 8 presents results on five tasks from the ImageNet-R benchmark. CaLoRA achieves the highest performance among all compared methods under the continual learning setting. These results support the applicability of CaLoRA across both NLP and vision tasks, demonstrating its generality in diverse continual learning scenarios.

**Analysis of Computation, Parameter, and Storage Efficiency.** As shown in Table 9, we compare the inference efficiency of various PEFT-based continual learning methods across three dimensions: GFLOPs, the number of trainable parameters, and feature storage requirements. Table 9 demonstrates that LoRA-based methods generally achieve superior inference efficiency. Compared with most state-of-the-art (SOTA) methods, our proposed CaLoRA does not significantly increase parameter or computational overhead while effectively improving continual learning performance.

