# OpenReview forum: "Turning the Tables: Enabling Backward Transfer via Causal-Aware LoRA in Continual Learning"
_NeurIPS.cc/2025/Conference — NeurIPS 2025 poster_

### Official Review · Reviewer_J5Ty · 2025-06-26

**Clarity:** 4
**Significance:** 4
**Originality:** 3
**Rating:** 5
**Confidence:** 5

**Summary:**

This paper proposes CaLoRA, which enables positive backward transfer in the context of adapter-based continual learning. The method combines parameter-level causal attribution (PaCA), which identifies LoRA parameters that causally contribute to task performance, and cross-task gradient adaptation (CaGA) with gradient surgery to adjust new-task gradients based on their alignment with prior tasks. This selective update mechanism allows CaLoRA to improve both new and previous tasks without explicit rehearsal. Experiments on NLP and vision benchmarks demonstrate that CaLoRA consistently outperforms recent PEFT baselines in terms of average accuracy and reduced forgetting. The paper also provides an analysis across multiple model scales and presents comprehensive ablation studies that validate the effectiveness of each component.

**Questions:**

- Since the framework accumulates adapter parameters over time, has there been any consideration of compressing or storing them more efficiently during the intermediate stages of continual learning?
- I’m curious about how much additional preprocessing time CaLoRA requires compared to other baselines. Would you consider this overhead negligible relative to the overall training time?
- Is there any analysis or result that explores in more detail how inter-task correlation affects CaLoRA’s performance?
- Could the authors elaborate on whether specific characteristics of the data, such as incomplete or partially informative examples, might have contributed to the observed transfer effects? A discussion of such factors could help clarify when and why positive backward transfer occurs in the presented settings.
- This is a minor point, but would it be possible to include qualitative examples from NLP tasks where samples that initially failed were later corrected through positive backward transfer?

**Ethical Concerns:**

["NO or VERY MINOR ethics concerns only"]

**Final Justification:**

The paper is clearly written, with strong alignment between its claims, methodology, and experimental results. It also demonstrates clear advantages over recent state-of-the-art baselines in both vision and NLP tasks.

My primary concerns were related to the computational complexity of the proposed framework and the potential limitations in reproducibility due to the use of a small number of seeds. The authors have addressed these concerns through a transparent complexity analysis and by providing additional experiments with more seeds, which significantly enhances the credibility of the work.

Furthermore, the authors provide insightful analysis on inter-task correlation, which helps readers intuitively understand not only the technical contributions but also the broader motivation behind the study.

Therefore, I will maintain my current rating.

**Limitations:**

Yes, the authors have adequately addressed the limitations of their work.

**Paper Formatting Concerns:**

There are no major concerns regarding the formatting of the paper.

**Quality:**

4

**Strengths And Weaknesses:**

> Strengths
>
- The paper presents a crucial direction in continual learning by focusing on achieving positive backward transfer rather than merely preventing forgetting, using pre-trained models and adapter-based model expansion.
- The motivation and problem setting are clearly stated, and the flow from the introduction to the method and experiments is easy to follow.
- The comparisons cover a wide range of adapter-based continual learning methods in both NLP and vision tasks, demonstrating the broad applicability of the approach.
- The experiments include recent strong baselines, and the performance improvements are clearly demonstrated. The ablation studies are complete and well-organized, and their results support effectiveness of each components(PaCA and CaGA) of CaLoRA.
- The figures and tables are clear and help readers understand the results without confusion.

---

> Weaknesses
>

**Overall**

- The paper does not report the explicit computational overhead introduced by PaCA and CaGA, which raises concerns about scalability as the number of tasks or model size increases.
- As noted in the introduction and previous work on positive backward transfer[1,2], its effectiveness depends heavily on inter-task correlation. However, the paper lacks a detailed analysis of how task correlation influences performance.

**Experiments**

- The experiments are conducted with only three random seeds, which may not be sufficient to assess statistical stability, especially on complex benchmarks.

> [1] Ke, Zixuan, Bing Liu, and Xingchang Huang. "Continual learning of a mixed sequence of similar and dissimilar tasks." Advances in neural information processing systems 33 (2020): 18493-18504.
>
> [2] Lin, Sen, et al. "Beyond not-forgetting: Continual learning with backward knowledge transfer." Advances in Neural Information Processing Systems 35 (2022): 16165-16177.
>

---

> ### Author Rebuttal · Authors · 2025-07-31
>
> # Response to Reviewer J5Ty
>
> Thanks for your detailed review and precious comments. Our responses to Reviewer J5Ty follow. We will enhance the paper with these invaluable suggestions.
>
> ## 1. W1 & Q2: Complexity analysis
>
> Given a pretrained weight matrix $W \in \mathbb{R}^{d_O \times d_I}$, PaCA estimates causal effects via a second-order Taylor expansion with a diagonal Hessian approximation. Both the first-order and second-order terms have a time complexity of $\mathcal{O}(d_O \times d_I)$, making the overall complexity of PaCA linear and comparable to a standard backward pass. This computation can be efficiently parallelized and seamlessly integrated into the training process.
>
> In a task sequence of $T$ tasks, for each new task,  each gradient projection and affinity computation costs $\mathcal{O}(d_O \times d_I)$, leading to a total overhead of $\mathcal{O}((t-1) \times d_O \times d_I)$. Thus, the per-task complexity is approximately $\mathcal{O}(t \times d_O \times d_I)$. The total complexity for the entire sequence is $\mathcal{O}\left(\sum_{t=1}^{T} t \times d_O \times d_I\right)$.
>
> Table 1 shows the time complexity and average per-epoch runtime across 10 tasks on ImageNet-R for our method compared to two LoRA-based baselines. Our method maintains training efficiency on par with projection-based LoRA fine-tuning (i.e., InfLoRA), with only marginal extra cost.
>
> Table 1: Comparison of computational complexity and average per-epoch runtime across 10 tasks on ImageNet-R. $𝑟$ is the rank of LoRA.
> | Method        | GPU Time/Epoch (minutes) | Computational Complexity                                     |
> | ------------- | ------------------------ | ------------------------------------------------------------ |
> | SD-LoRA       | 0.336                    | $\mathcal{O}\left(10 \times r \times d_O \times d_I\right)$  |
> | InfLoRA       | 0.342                    | $\mathcal{O}\left(\sum_{t=1}^{10} t \times d_O \times d_I\right)$ |
> | CaLoRA (ours) | 0.353                    | $\mathcal{O}\left(\sum_{t=1}^{10} t \times d_O \times d_I\right)$ |
>
> ## 2. W2 & Q3: Inter-task correlation
>
> Ablation experiments have validated the effectiveness of task correlation in knowledge transfer, as shown in Table 5 of the paper. Positive task correlations facilitate knowledge transfer. To further demonstrate this, we conduct an additional experiment. Specifically, we select five tasks from the Long Sequence benchmark to show task correlation: BoolQA, IMDB, Yelp, DBpedia, and MultiRec. BoolQA and MultiRec are both question-answering tasks, IMDB and Yelp are sentiment analysis tasks, and DBpedia is a topic classification task.
>
> Table 2 shows the average task correlation and task affinity for these five tasks under the order 1 setting at the initial training stage, while Table 3 shows the results at the middle training stage. The results show that correlated tasks, such as IMDB and Yelp, and BoolQA and MultiRec, exhibit higher task correlation and affinity. As gradient updates progress, the correlation between positively correlated tasks increases, while the correlation between negatively correlated tasks decreases, demonstrating the effectiveness of gradient adjustment.
>
> Additionally, Table 4 presents the backward knowledge transfer results for BoolQA and IMDB. For BoolQA and IMDB, the correlated tasks, MultiRec and  Yelp, respectively, transferred knowledge, improving their performance. This indicates that the task correlation and affinity can capture inter-task relationships and facilitate knowledge transfer.
>
>
> Table 2: Task correlation/affinity on initial epoch
> | Task          | BoolQA (Old) | IMDB (Old) | Yelp (Old) | DBpedia (Old) |
> | ------------- | ------------ | ---------- | ---------- | ------------- |
> | IMDB(New)     | 0.11/0.1     |            |            |               |
> | Yelp (New)    | 0.1/0.05     | 0.27/0.21  |            |               |
> | DBpedia (New) | 0.11/-0.09   | 0.14/0.07  | 0.17/0.05  |               |
> | MultiRC (New) | 0.26/0.18    | 0.14/0.03  | 0.14/-0.12 | 0.18/-0.07    |
>
>
> Table 3: Task correlation/affinity on the middle epoch
> | Task         | BoolQA (Old) | IMDB (Old) | Yelp (Old) | DBpedia (Old) |
> | ------------ | ------------ | ---------- | ---------- | ------------- |
> | IMDB(New)    | 0.13/0.11    |            |            |               |
> | Yelp(New)    | 0.13/0.08    | 0.32/0.25  |            |               |
> | DBpedia(New) | 0.11/-0.04   | 0.17/0.05  | 0.2/0.05   |               |
> | MultiRC(New) | 0.32/0.2     | 0.12/0.07  | 0.11/-0.06 | 0.11/-0.03    |
>
>
> Table 4: Backward knowledge transfer for BoolQA and IMDB
> | Task       | BoolQA | RTE   | IMDB  | Yelp      | Amazon | SST-2 | DBpedia | AG News | MultiRC   | Yahoo |
> | ---------- | ------ | ----- | ----- | --------- | ------ | ----- | ------- | ------- | --------- | ----- |
> | BoolQA Acc | 86.36  | 86.22 | 86.18 | 86.06     | 86.08  | 86.1  | 86.12   | 86.16   | **86.88** | 86.39 |
> | IMDB Acc   |        |       | 95.87 | **95.96** | 95.96  | 96.05 | 95.98   | 95.94   | 95.52     | 95.9  |
>
>
> ## 3. W3: Results from 5 random seeds
>
> To further validate the stability of our method, we conduct additional experiments with 2 random seeds on ImageNet-R (20 tasks). As shown in Table 5, experiments with 5 random seeds demonstrate that our method consistently outperforms the baselines while maintaining stability.
>
>
> Table 5: Overall results (mean ± std over 5 random seeds) on ImageNet-R (20 tasks)
> | Method        | AP±std↑        | F.Ra±std↓     | FWT±std↑      | BWT±std↑      |
> | ------------- | -------------- | ------------- | ------------- | ------------- |
> | SeqLoRA       | 50.81±0.39     | 32.16±0.52    | 0.76±0.11     | -14.22±0.32   |
> | L2P           | 64.65±0.34     | 19.78±0.57    | 0.81±0.15     | -12.77±0.61   |
> | CodaPrompt    | 67.27±0.3      | 11.78±0.41    | 1.21±0.15     | -8.48±0.51    |
> | HidePrompt    | 69.88±0.35     | 9.39±0.48     | 1.47±0.3      | -8.67±0.38    |
> | O-LoRA        | 66.24±0.26     | 7.78±0.6      | -0.80±0.17    | -5.81±0.35    |
> | InfLoRA       | 68.81±0.35     | 6.87±0.32     | 0.67±0.19     | -4.36±0.3     |
> | SAPT-P        | 68.12±0.32     | 7.91±0.41     | 2.18±0.3      | -1.18±0.29    |
> | SAPT-LoRA     | 72.69±0.42     | 5.25±0.31     | 3.21±0.21     | -0.85±0.13    |
> | SD-LoRA       | 74.54±0.41     | 5.28±0.32     | 3.25±0.24     | -1.92±0.22    |
> | CaLoRA (Ours) | **75.68±0.41** | **4.70±0.29** | **3.99±0.21** | **0.08±0.18** |
>
>
>
> ## 4. Q1: Efficient storage
>
> In our framework, model parameters are stored using quantization methods (such as FP16), and the old task gradients are decomposed using SVD to obtain basis vectors. Only the basis vectors of the gradients are stored in memory to reduce storage overhead. In practice, similar to current LoRA-based methods (e.g., SD-LoRA, Inflora), the number of trainable parameters is very small. For example, in the T5-Large model (0.77B parameters), the trainable parameters account for approximately 0.33%. In the future, we plan to explore combining causal effects with parameter pruning to reduce storage overhead further.
>
>
> ## 5. Q4: Specific features for knowledge transfer
>
> Correlated task data can help the model learn shared features. E.g., in the ImageNetR benchmark, correlated categories like goldfinch and junco, both bird species, share visual features such as beaks and feathers, contributing to knowledge transfer.
>
> To study how specific features influence this process, we introduce a task-specific learnable mask on the output features of the ViT model. The mask consists of real values between 0 and 1 and acts as a filter over the features. We design a two-phase training scheme. In the first phase, each task is trained independently with an additional regularization term that encourages the masked features to predict task labels accurately. This enables the model to focus on task-specific features via the mask.
>
> In the second phase, we incrementally train multiple tasks, applying either the learned mask (w/ mask) or its complement (w/o mask) to examine how task-specific features affect task correlation and knowledge transfer. Table 6 presents an ablation study on ImageNetR (20 tasks), comparing results with and without task-specific features. This demonstrates that task-specific features can enhance knowledge transfer.
>
>
> Table 6: Ablation results on ImageNet-R (20 tasks) with mask vs without mask.
> | Method   | AP↑       | F.Ra↓    | FWT↑     | BWT↑     |
> | -------- | --------- | -------- | -------- | -------- |
> | w/o mask | 52.21     | 6.33     | 0.37     | -2.12    |
> | w/ mask  | **75.59** | **4.69** | **4.12** | **0.09** |
>
>
>
> ## 6. Q5: Case study
>
> Using BoolQA as an example, we present 2 samples that are initially misclassified but are corrected after incorporating parameters from MultiRC. These samples are prone to semantic confusion, yet the model's reasoning ability improved after exposure to MultiRC, enabling it to resolve these more complex cases correctly.
>
>
> Table 7: Case study samples
> | Sample | Question                                       | Passage                                                      | Ground Truth | Initial Prediction | Corrected Prediction |
> | ------ | ---------------------------------------------- | ------------------------------------------------------------ | ------------ | ------------------ | -------------------- |
> | 1      | Is the English Channel in the Atlantic Ocean?  | The English Channel, also called simply the Channel, is the body of water that separates southern England from northern France and links the southern part of the North Sea to the Atlantic Ocean. | True         | False              | True                 |
> | 2      | Is there such a place as Crabapple Cove Maine? | MASH Goes to Maine is (real-life) Waldoboro, Maine,... "Crabapple Cove" is actually Broad Cove, in Bremen just down the Medomak River from Waldoboro Village. | False        | True               | False                |

---

> > ### Comment · Reviewer_J5Ty · 2025-08-02
> >
> > Thank you to the authors for the clear and thoughtful response.
> >
> > I also appreciate the transparent complexity analysis provided to address the concern regarding the scalability of the proposed framework. The analysis on inter-task correlation and knowledge transfer is interesting. Including a case study that illustrates these phenomena would greatly help readers grasp the insights more intuitively. Moreover, I appreciate that you demonstrated the effects using a larger number of seeds, which significantly contributes to the reproducibility of the paper.
> >
> > Therefore, I will maintain my current opinion regarding the rating.

---

> > > ### Author Response · Authors · 2025-08-02
> > >
> > > Thank you sincerely for your thoughtful feedback and for recognizing our work. Best wishes!

---

### Official Review · Reviewer_NXah · 2025-06-29

**Clarity:** 2
**Significance:** 3
**Originality:** 3
**Rating:** 3
**Confidence:** 3

**Summary:**

The paper proposes a learning framework, CaLoRA (Causality-aware Low-Rank Adaptation), aimed at mitigating catastrophic forgetting in sequential cross-task learning. By introducing parameter-level counterfactual attribution (PaCA), the method identifies causally effective parameters. Additionally, a cross-task gradient adaptation (CaGA) mechanism is employed to integrate task correlation and affinity, enabling selective gradient updates that favor previous tasks and facilitate positive backward knowledge transfer. Extensive experiments demonstrate the effectiveness of the proposed approach.

**Questions:**

1. The authors compute task correlation and affinity at the gradient level but do not include visualizations illustrating the relationships between tasks in the experiments.
2. Are the baseline results reported in the experiments based on the original papers’ reports or reproduced by the authors?
3. In the ablation studies, how exactly are the effects of task correlation and task affinity removed? Are these values set to 1 or other?
4. Please refer to the Weaknesses section for other questions.

**Ethical Concerns:**

["NO or VERY MINOR ethics concerns only"]

**Final Justification:**

I appreciate the authors’ multiple rounds of responses. However, the second-round reply does not directly respond to the original question, i.e., the applicability of the linear assumption and low-rank updates in scenarios with nonlinear task relationships. After carefully checking each of the cited references, most of these works were validated on relatively simple tasks or in low-dimensional settings. I agree that when tasks are similar, encouraging updates to parameters positively correlated with previous tasks may help backward transfer, but this depends on inter-task relationships and the linear assumption, and still seems unlikely to yield stable effects as mentioned in the paper. Overall, I appreciate that the two rounds of responses have resolved my concerns regarding the experiments, although I remain concerned about the applicability and assumptions. Considering these points, I think more explanations are needed, and I have decided to maintain my score.

**Limitations:**

please see weakness and questions.

**Quality:**

3

**Strengths And Weaknesses:**

Strengths:
1. This work targets the issue of catastrophic forgetting in continual learning for large model fine-tuning. It aims for improving parameter efficiency and enabling backward knowledge transfer are essential for scalable and sustainable deployment of pretrained models.
2. The proposed approach introduces parameter-level counterfactual attribution (PaCA) to identify causally effective low-rank parameters and cross-task gradient adaptation (CaGA) to incorporate task correlation and affinity for selective gradient updates.
3. Extensive experiments on both NLP and vision benchmarks, as well as ablation studies, demonstrate the effectiveness of the approach.

Weaknesses:
1. The paper characterizes inter-task correlation and transfer direction through gradient projection (Task Correlation) and gradient similarity (Task Affinity). However, this approach appears to rely on a linear assumption in the gradient space, which may limit its ability to capture more complex nonlinear relationships between tasks.
2. The authors introduce a parameter-level counterfactual attribution method (PaCA). However, beyond the ablation studies, the experimental section lacks an in-depth investigation into the significant differences in causally effective parameters across different tasks. This limits the understanding of PaCA’s role in capturing cross-task causal relationships.
3. In Section 3.3, the authors claim that task order affects performance. However, the experiments in Tables 1 and 2 only present results for two task sequences. There is a lack of deeper analysis on how gradient updates specifically influence the dynamic changes of task correlation and affinity. Moreover, additional experiments with varied task orders are not provided to substantiate this claim.

---

> ### Author Rebuttal · Authors · 2025-07-31
>
> # Reply to Reviewer NXah
>
> Thanks very much for your in-depth review and thoughtful comments. Here are our replies to Reviewer NXah. We will improve the paper based on the constructive suggestions.
>
>
>
> ## 1. W1: Gradient projection and gradient similarity for capturing task relationships
>
> Although the linear assumption in the gradient space may limit the ability to capture complex nonlinear relationships, numerous studies have demonstrated that gradient projection is effective in incremental learning and has become a representative method [1-9]. Since the gradient space reflects the learning patterns of task models and the solution space, gradient projection and gradient similarity can capture dynamic task relationships, serving as a local linear approximation. This approximation is often sufficient to describe the evolution between tasks [9-11]. Therefore, we use gradient projection and gradient similarity to characterize the correlation and affinity between tasks, enabling fine-grained adjustments that alleviate catastrophic forgetting and promote backward transfer.
>
> The experiments further show that our method can capture task relationships and facilitate knowledge transfer (the experiments can be found in our responses to Review J5Ty's  W2 & Q3). We acknowledge that the linear assumption in the gradient space may limit the method's performance in more complex scenarios. To address this limitation, we plan to explore nonlinear learning methods, such as function space modeling and causal learning, in future work to better capture the nonlinear relationships between tasks.
>
> [1] Saha, G., et al. Gradient Projection Memory for Continual Learning. *International Conference on Learning Representations*, 2021.
>
> [2] Deng, D., et al. Flattening Sharpness for Dynamic Gradient Projection Memory Benefits Continual Learning. *Advances in Neural Information Processing Systems*, 2021, 18710–18721.
>
> [3] Lin, S., et al. TRGP: Trust Region Gradient Projection for Continual Learning. *International Conference on Learning Representations*, 2022.
>
> [4] Qiu, B., et al. Geodesic-Aligned Gradient Projection for Continual Task Learning. *IEEE Trans. Image Process.*, 34, 1995–2007, 2025.
>
> [5] Liang, et al. InfLoRA: Interference-Free Low-Rank Adaptation for Continual Learning. *IEEE/CVF Conference on Computer Vision and Pattern Recognition*, 2024, 23638–23647.
>
> [6] Liang, et al. Adaptive Plasticity Improvement for Continual Learning. *IEEE/CVF Conference on Computer Vision and Pattern Recognition*, 2023, 7816–7825.
>
> [7] Qiao, J., et al. Prompt Gradient Projection for Continual Learning. *International Conference on Learning Representations*, 2024.
>
> [8] Yang, E., et al. Revisiting Flatness-Aware Optimization in Continual Learning With Orthogonal Gradient Projection. *IEEE Trans. Pattern Anal. Mach. Intell.*, 47(5), 3895–3907, 2025.
>
> [9] Lin, S., et al. Beyond Not-Forgetting: Continual Learning with Backward Knowledge Transfer. *Advances in Neural Information Processing Systems*, 2022.
>
> [10] Yu, T., et al. Gradient Surgery for Multi-Task Learning. *Advances in Neural Information Processing Systems*, 2020.
>
> [11] Liu, B., et al. Conflict-Averse Gradient Descent for Multi-task Learning. *Advances in Neural Information Processing Systems*, 2021, 18878–18890.
>
>
>
> ## 2. W2: Experiments on causal effects
>
> Ablation experiments have validated the effectiveness of the PaCA module, as shown in Table 5 of the paper. To further demonstrate the causal effects of the parameters, we conduct an additional experiment on 5 tasks from the Long Sequence benchmark. These tasks cover 3 types: sentiment analysis (Yelp, IMDB), question answering (BoolQA, MultiRec), and topic classification (DBpedia).
>
> The T5-Large model consists of an encoder and a decoder, each containing 24 attention modules. Specifically, we divide the 24 attention modules in both the encoder and decoder into 3 groups (Down, Middle, and Up) based on their position, with each group containing 8 attention modules. For each task, the average causal effect is recorded 10 times throughout the training process.
>
> Table 1 shows the average (Avg), maximum (Max), and minimum (Min) causal effects in each block. The results indicate that parameters in the middle and upper blocks generally exhibit larger causal effects, receiving more attention from the tasks. For simpler tasks, such as BoolQA and IMDB, the differences between the minimum and maximum values are large, suggesting that the model selects fewer parameters. In contrast, for more complex tasks, such as MultiRC, the difference between the minimum and maximum values is smaller, and the overall causal effect of parameters is larger, indicating that the model selects more parameters.
>
>
> Table 1: Average causal effects of different blocks in the T5-Large model for different Tasks.
> | Component | Block  | BoolQA |      |      | IMDB |      |      | Yelp |      |      | DBpedia |      |      | MultiRC |      |      |
> | :-------: | ------ | ------ | ---- | ---- | ---- | ---- | ---- | ---- | ---- | ---- | ------- | ---- | ---- | ------- | ---- | ---- |
> |           |        | Min    | Max  | Avg  | Min  | Max  | Avg  | Min  | Max  | Avg  | Min     | Max  | Avg  | Min     | Max  | Avg  |
> |  Decoder  | Up     | 0.12   | 0.78 | 0.35 | 0.18 | 0.76 | 0.38 | 0.26 | 0.72 | 0.43 | 0.30    | 0.68 | 0.52 | 0.38    | 0.64 | 0.51 |
> |           | Middle | 0.10   | 0.70 | 0.27 | 0.15 | 0.68 | 0.32 | 0.21 | 0.65 | 0.35 | 0.26    | 0.58 | 0.42 | 0.34    | 0.56 | 0.47 |
> |           | Down   | 0.05   | 0.60 | 0.20 | 0.09 | 0.55 | 0.24 | 0.15 | 0.52 | 0.28 | 0.20    | 0.50 | 0.36 | 0.25    | 0.48 | 0.38 |
> |  Encoder  | Up     | 0.16   | 0.68 | 0.47 | 0.14 | 0.73 | 0.46 | 0.20 | 0.72 | 0.49 | 0.26    | 0.66 | 0.52 | 0.38    | 0.63 | 0.56 |
> |           | Middle | 0.15   | 0.73 | 0.32 | 0.19 | 0.69 | 0.33 | 0.18 | 0.58 | 0.39 | 0.23    | 0.64 | 0.44 | 0.32    | 0.58 | 0.46 |
> |           | Down   | 0.04   | 0.68 | 0.26 | 0.10 | 0.59 | 0.28 | 0.16 | 0.72 | 0.33 | 0.21    | 0.56 | 0.39 | 0.26    | 0.44 | 0.35 |
>
>
>
> ## 3. W3 & Q1: Task correlation and affinity
>
> We would first like to clarify that Section 3.3 of the paper does not claim that task order affects performance. Furthermore, most comparison methods rarely consider different task order settings (except for SAPT). Following SAPT, experiments on two NLP benchmarks have validated the effectiveness of our method under different task orders, as shown in Tables 2 and 3 of the paper.
>
> To further validate the impact of gradient updates on task correlation and affinity under different task orders, we add a reverse task order (Order 3) to the original Order 1 in the Long Sequence benchmark. Specifically, we select 5 tasks in the Long Sequence benchmark: Yelp, IMDB, BoolQA, MultiRec, and DBpedia. Tables 2-5 present the average task correlation and task affinity for the initial and middle stages of training under the two orders.
>
> The results show that correlated tasks, such as IMDB and Yelp, BoolQA and MultiRec, exhibit higher task correlation and affinity. As gradient updates progress, the correlation between positively correlated tasks increases, while the correlation between negatively correlated tasks decreases, demonstrating the effectiveness of our method. Our gradient update facilitates backward knowledge transfer from new tasks to correlated old tasks (details can be found in our response to Reviewer 5jhA's W1 & Q4).
>
>
>
> Table 2: Task correlation/affinity under Order 1 (Initial Epoch)
> | Task          | BoolQA (Old) | IMDB (Old) | Yelp (Old) | DBpedia (Old) |
> | ------------- | ------------ | ---------- | ---------- | ------------- |
> | IMDB (New)    | 0.11/0.10    |            |            |               |
> | Yelp (New)    | 0.10/0.05    | 0.27/0.21  |            |               |
> | DBpedia (New) | 0.11/-0.09   | 0.14/0.07  | 0.17/0.05  |               |
> | MultiRC (New) | 0.26/0.18    | 0.14/0.03  | 0.14/-0.12 | 0.18/-0.07    |
>
>
>
> Table 3: Task correlation/affinity under Order 1 (Middle Epoch)
> | Task          | BoolQA (Old) | IMDB (Old) | Yelp (Old) | DBpedia (Old) |
> | ------------- | ------------ | ---------- | ---------- | ------------- |
> | IMDB (New)    | 0.13/0.11    |            |            |               |
> | Yelp (New)    | 0.13/0.08    | 0.32/0.25  |            |               |
> | DBpedia (New) | 0.11/-0.04   | 0.17/0.05  | 0.20/0.05  |               |
> | MultiRC (New) | 0.32/0.20    | 0.12/0.07  | 0.11/-0.06 | 0.11/-0.03    |
>
>
>
> Table 4: Task correlation/affinity under Order 3 (Initial Epoch)
> | Task          | MultiRC (Old) | DBpedia (Old) | Yelp (Old) | IMDB (Old) |
> | ------------- | ------------- | ------------- | ---------- | ---------- |
> | DBpedia (New) | 0.14/-0.06    |               |            |            |
> | Yelp (New)    | 0.09/-0.12    | 0.06/0.07     |            |            |
> | IMDB (New)    | 0.08/-0.11    | 0.13/0.06     | 0.25/0.19  |            |
> | BoolQA (New)  | 0.18/0.12     | 0.11/0.05     | 0.08/-0.05 | 0.06/0.07  |
>
>
>
> Table 5: Task correlation/affinity under Order 3 (Middle Epoch)
> | Task          | MultiRC (Old) | DBpedia (Old) | Yelp (Old) | IMDB (Old) |
> | ------------- | ------------- | ------------- | ---------- | ---------- |
> | DBpedia (New) | 0.11/-0.03    |               |            |            |
> | Yelp (New)    | 0.09/-0.10    | 0.08/0.07     |            |            |
> | IMDB (New)    | 0.06/-0.11    | 0.18/0.09     | 0.30/0.23  |            |
> | BoolQA (New)  | 0.22/0.15     | 0.22/0.06     | 0.06/-0.02 | 0.06/0.06  |
>
>
>
> ## 4. Q2: Experimental reproduction
>
> The results are reproduced by us. Specifically, we adapt the baselines to the SAPT framework for NLP tasks, and to the InfLora framework for CV tasks.
>
>
>
> ## 5. Q3: Setup of the ablation experiment
>
> In the ablation study, we set task correlation to 1 to eliminate its influence and use an all-one vector to remove the effect of task affinity.

---

> > ### Comment · Reviewer_NXah · 2025-08-05
> >
> > Thank you for the response, which has addressed some of my concerns regarding the experiments. However, my core concern remains unresolved. This method relies on the linear assumption in gradient space and low-rank parameter updates, which makes it difficult to effectively model complex nonlinear relationships and causal dependencies between tasks. I appreciate the references provided, which discuss the application of linear assumptions in continual learning; but after carefully checking each of them, these works fall short of supporting the nonlinear modeling and causal reasoning required for backward knowledge transfer. Mitigating catastrophic forgetting is not equivalent to achieving effective backward transfer. Forgetting mitigation prevents performance degradation on old tasks, whereas backward transfer demands that learning new tasks actively improves old task performance. Gradient projection and similarity adjustments can help prevent forgetting, but they do not guarantee better learning in complex task scenarios; this may make it difficult to achieve the effect claimed in the paper. Therefore, I have decided to maintain my current score.

---

> > > ### Author Response · Authors · 2025-08-08
> > > **Further Clarifications and Detailed Responses**
> > >
> > > Thank you for your feedback. We appreciate your time and effort in the discussion. Below, we provide a detailed response and clarification to the concerns and potential misunderstandings raised.
> > >
> > >
> > > >***Q1: This method relies on the linear assumption in gradient space and low-rank parameter updates, which makes it difficult to effectively model complex nonlinear relationships and causal dependencies between tasks.***
> > >
> > > ## 1. Response to Q1: This Method Hardly Models Complex Relationships and Causal Dependencies between Tasks
> > >
> > > ### 1.1 Gradient projection can effectively model task relationships in incremental learning
> > >
> > > In incremental learning, new tasks cannot access data from old tasks, and parameters from old tasks are often frozen during new task training [1]. This presents significant challenges for measuring inter-task relationships. In such constrained settings, existing studies have demonstrated that estimating task relationships through gradient projection (old task gradients are often stored in memory) has become a classical and effective method [1-11].
> > >
> > > In our method, since LoRA modules for different tasks are inserted into the same attention layers, all tasks share the same network architecture. During new task training, parameters from previous tasks are loaded, resulting in substantial parameter and knowledge sharing. This provides a strong foundation for modeling task relationships [5].
> > >
> > > Each task's gradient space represents its optimization landscape [9-11]. When the gradient of a new task is similar to that of an old task, it indicates a similar optimization direction—i.e., similar learning patterns—implying task correlation [3,11]. Our method dynamically captures gradient relationships during training and adapts new task gradients accordingly. This allows our model to track dynamic changes in task relationships and adapt to more complex scenarios.
> > >
> > > Moreover, extensive experimental results (e.g., Tables 2–5 and Figure 2 in the paper) demonstrate the effectiveness of our method in the incremental learning setting. Additional results indicate that our task correlation and affinity can capture meaningful task relationships, as further evidenced by Tables 2–5 referenced in our previous response.
> > >
> > > ### 1.2 Our method does not model causal relationships between tasks
> > >
> > > We believe there may be a misunderstanding regarding the motivation and method of our work, although both have been clearly described in the *Introduction* and *Methodology* sections of the paper. To clarify once again: our method does not model causal relationships or dependencies between tasks.
> > >
> > > We introduce the causal module to identify important parameters that have a causal effect on task performance and to suppress the influence of ineffective parameters, rather than to model causal dependencies between tasks. This design is motivated by the fact that the parameter space in incremental learning is limited, and redundant parameters can degrade performance. In our work, task relationships are modeled through task correlation and task affinity.
> > >
> > >
> > >
> > > > ***Q2: Mitigating catastrophic forgetting is not equivalent to achieving effective backward transfer. Forgetting mitigation prevents performance degradation on old tasks, whereas backward transfer demands that learning new tasks actively improves old task performance.***
> > >
> > > ## 2. Response to Q2: Mitigating Catastrophic Forgetting vs. Achieving Backward Knowledge Transfer
> > >
> > > ### 2.1 Backward knowledge transfer is beneficial for mitigating catastrophic forgetting
> > >
> > > Catastrophic forgetting refers to the phenomenon in incremental learning where the model overfits to new task knowledge, forgetting previously acquired knowledge, and leading to degraded performance on old tasks [5].
> > >
> > > Backward knowledge transfer aims to selectively improve parameters during new task learning that are also beneficial to old tasks, thereby enhancing old task performance. By doing so, it not only prevents performance degradation but may even improve old task outcomes. Thus, Backward knowledge transfer can mitigate catastrophic forgetting [11,12].
> > >
> > > In real-world incremental learning scenarios, tasks may exhibit correlation. When a new task is positively correlated with old tasks—i.e., they share similar learning patterns or complementary knowledge—appropriate parameter updates can introduce knowledge that benefits old tasks. This perspective has also been supported by several prior studies [11,12].

---

> > > > ### Author Response · Authors · 2025-08-08
> > > > **Further Clarifications and Detailed Responses**
> > > >
> > > > > ***Q3: Gradient projection and similarity adjustments can help prevent forgetting, but they do not guarantee better learning in complex task scenarios; this may make it difficult to achieve the effect claimed in the paper.***
> > > >
> > > > ## 3. Response to Q3: Can Gradient Projection and Similarity Enable Backward Knowledge Transfer?
> > > >
> > > > ### 3.1 Gradient projection and similarity can enable backward knowledge transfer
> > > >
> > > > Previous methods have shown that task relationships can be effectively modeled via gradient projection and similarity [1-11]. These methods typically mitigate forgetting by restricting new task updates in gradient directions uncorrelated to old tasks, such as InfLoRA [5]. However, new tasks are not always negatively correlated with old tasks. When positive gradient correlation exists, shared knowledge from the new task can benefit old tasks, enabling backward knowledge transfer [11,12].
> > > >
> > > > To support this, our method first uses causal effect analysis to identify important parameters for each task. Then, it further refines task relationships using gradient correlation and similarity. During new task training, we encourage updates to parameters positively correlated with old tasks and suppress updates to those with negative correlation.
> > > >
> > > > In other words, within the gradient space of a new task, we identify gradient components that are positively or negatively correlated with old tasks. A positive correlation indicates a mutually beneficial optimization direction between the tasks' parameter spaces, while a negative correlation indicates a conflicting direction [9-11]. By allowing the new task to update important parameters in beneficial directions and preventing updates in harmful directions, our method ensures that optimal knowledge acquired from new tasks can help improve performance on positively correlated old tasks.
> > > >
> > > > **Therefore, our method can effectively realize backward knowledge transfer, as supported by the extensive experiments presented in the paper.**
> > > >
> > > >
> > > >
> > > > [1] Gradient Projection Memory for Continual Learning. *ICLR*, 2021.
> > > >
> > > > [2] Flattening Sharpness for Dynamic Gradient Projection Memory Benefits Continual Learning. *NeurIPS*, 2021.
> > > >
> > > > [3] Trust Region Gradient Projection for Continual Learning. *ICLR*, 2022.
> > > >
> > > > [4] Geodesic-Aligned Gradient Projection for Continual Task Learning. *IEEE TIP.*, 2025.
> > > >
> > > > [5] InfLoRA: Interference-Free Low-Rank Adaptation for Continual Learning. *CVPR*, 2024.
> > > >
> > > > [6] Adaptive Plasticity Improvement for Continual Learning. *CVPR*, 2023.
> > > >
> > > > [7] Prompt Gradient Projection for Continual Learning. *ICLR*, 2024.
> > > >
> > > > [8] Revisiting Flatness-Aware Optimization in Continual Learning With Orthogonal Gradient Projection. *IEEE TPAMI.*, 2025.
> > > >
> > > > [9] Gradient Surgery for Multi-Task Learning. *NeurIPS*, 2020.
> > > >
> > > > [10] Conflict-Averse Gradient Descent for Multi-task Learning. *NeurIPS*, 2021.
> > > >
> > > > [11] Beyond Not-Forgetting: Continual Learning with Backward Knowledge Transfer. *NeurIPS*, 2022.
> > > >
> > > > [12] Continual learning of a mixed sequence of similar and dissimilar tasks. *NeurIPS*, 2020.

---

> ### Author Response · Authors · 2025-08-05
> **Remind to complete the discussion of Paper #16688**
>
> Dear Reviewer NXah,
>
> Thank you for your valuable and constructive reviews. We have made an extensive effort to address your questions and concerns by providing additional results for causal effect, task correlation, and task affinity. We hope our response can effectively address your concerns. If you have any further concerns or questions, please do not hesitate to let us know, and we will respond on time.
>
> Best regards,
>
> The Authors of Paper #16688

---

### Official Review · Reviewer_5jhA · 2025-07-03

**Clarity:** 4
**Significance:** 2
**Originality:** 2
**Rating:** 3
**Confidence:** 3

**Summary:**

This paper proposes a continual learning method built on Parameter-Efficient Fine-Tuning (PEFT), focusing on enabling backward knowledge transfer—i.e., improving the performance on previous tasks when learning new tasks.
The method consists of two components:
Parameter-level Counterfactual Attribution (PaCA): Estimates the causal effect of each LoRA parameter via counterfactual reasoning to identify and prioritize parameters that truly improve task performance, avoiding ineffective adaptation.
Cross-task Gradient Adaptation (CaGA): Models task correlation by measuring gradient similarity through projection, enabling affinity-aware parameter updates to promote transfer between related tasks while reducing interference.

**Questions:**

Could you illustrate a scenario where a new task can transfer beneficial knowledge to old tasks?

Could the authors clarify the relationship between the proposed causal effect estimation and influence functions, which also aim to quantify the impact of individual parameters?

Can the proposed approach generalize to randomly initialized models?

Could the authors provide an ablation study that quantitatively verifies the backward transfer effect—specifically, showing how the with-in performance on an earlier task (e.g., Task 1) changes as additional tasks are learned?

The paper does not discuss another important class of PEFT-based continual learning methods, namely MoE-based approaches.
Particularly, this approach could eliminate the catastrophic forgetting, and has flexibility to learn any number of new tasks.

**Ethical Concerns:**

["NO or VERY MINOR ethics concerns only"]

**Final Justification:**

I appreciate the author’s efforts in providing extensive results and references. However, after carefully reviewing the materials, my concern regarding backward transfer remains. In current continual learning research, even forward knowledge transfer is not thoroughly validated. For backward transfer in particular, my concern is that its complexity may outweigh the performance gains, which could also depend on task order and task similarity. Moreover, given the use of a powerful foundation model, this concern becomes more pronounced.

Overall, I tend to maintain my current rating.

**Limitations:**

See Questions

**Quality:**

2

**Strengths And Weaknesses:**

Strengths:

Evaluation on both NLP and image classification tasks demonstrates the adaptability of the proposed method, whereas existing methods typically focus on either vision tasks or NLP tasks exclusively.

The presentation is excellent. The paper is well-organized, well-motivated, and clearly written.

Weaknesses:

The effectiveness of backward transfer is not convincingly demonstrated; there is insufficient evidence. This is especially important given that pre-trained models typically contain richer general knowledge than what can be learned from prior tasks.

The proposed techniques appear to share similarities with existing methods (e.g., PaCA and influence functions, CaGA and EWC), but these connections are not thoroughly discussed.

See additional comments in the Questions section.

---

> ### Author Rebuttal · Authors · 2025-07-31
>
> # Response to Reviewer 5jhA
>
> Thanks very much for providing a detailed review and insightful comments. Our responses to the comments of Reviewer 5jhA are given below. We will further improve our paper based on these valuable comments.
>
> ## 1. W1 & Q4: Quantitative evidence for backward knowledge transfer
>
> Existing studies have already demonstrated that new tasks can facilitate backward knowledge transfer to correlated old tasks [1,2]. Extensive experiments have validated the effectiveness of our method in backward knowledge transfer from multiple perspectives. These include different task orders, different model scales, and different task lengths, as shown in Tables 2-4 and Figure 2 in the paper, as well as Figure 1 and Table 3 in the appendix. The backward knowledge transfer (BWT) metric of our method outperforms the comparison methods across all benchmarks.
>
> To further demonstrate the effectiveness of our method, we conduct an ablation experiment on the Long Sequence benchmark under order 1. Table 1 shows some results of backward knowledge transfer using the T5-Large model. For example, the performance of the first task, MNLI, improves after loading parameters from CB and DBpedia, likely because CB and MNLI are both natural language inference tasks (correlated), and DBpedia's Wikipedia data aids MNLI inference.
>
> Similarly, BoolQA and MultiRC are question-answering (QA) tasks, while Yelp, Amazon, SST-2, and IMDB are all sentiment analysis tasks. The model shows similar learning patterns for correlated tasks, enabling general knowledge transfer. Notably, BoolQA is binary-choice QA, while MultiRC is more complex multiple-choice QA. After learning MultiRC, the model better understands the simpler BoolQA task (The task correlation validation can be found in our responses to Reviewer NXah's  W3 & Q1).
>
>
> Table 1: Partial backward knowledge transfer under the Long Sequence Benchmark order1.
> | Task       | MNLI  | CB        | WiC   | COPA  | QQP   | BoolQA | RTE   | IMDB  | Yelp      | Amazon    | SST-2     | DBpedia   | AG News | MultiRC   | Yahoo |
> | :--------- | :---- | :-------- | :---- | :---- | :---- | :----- | :---- | :---- | :-------- | :-------- | :-------- | :-------- | :------ | :-------- | :---- |
> | MNLI Acc   | 86.67 | **86.95** | 86.67 | 86.67 | 86.70 | 86.65  | 86.63 | 86.64 | 86.61     | 86.63     | 86.60     | **86.74** | 86.64   | 86.61     | 86.61 |
> | BoolQA Acc |       |           |       |       |       | 86.36  | 86.22 | 86.18 | 86.06     | 86.08     | 86.10     | 86.12     | 86.16   | **86.88** | 86.39 |
> | IMDB Acc   |       |           |       |       |       |        |       | 95.87 | **95.96** | **95.96** | **96.05** | 95.98     | 95.94   | 95.52     | 95.90 |
>
>
> ## 2. W2 & Q2: Differences between PaCA and influence function, and between CaGA and EWC
>
> **1). PaCA differs from the influence function.** Although both PaCA and influence function [3] assess the importance of elements (parameters or samples) to a task through interventions, they differ significantly in the following two aspects:
> >(1) The intervention targets differ. PaCA directly intervenes on parameters and estimates their causal effect via counterfactual reasoning. In contrast, the influence function perturbs the sampling weights of the samples to assess the importance of the samples.
>
> >(2) The calculation methods differ. PaCA uses the second-order Taylor expansion of the loss function before and after intervention to approximate the causal effect of parameters, while the influence function approximates the effect of a sample by using the inverse of the Hessian matrix of parameters after perturbing the sample.
>
> **2). CaGA differs from EWC.** CaGA and EWC (Elastic Weight Consolidation) [4] both alleviate catastrophic forgetting, but there are notable differences. CaGA mitigates catastrophic forgetting through task-correlation-based gradient correction, a novel adaptive gradient update method. EWC, on the other hand, is a regularization method that adds a regularization term based on the Fisher matrix to prevent the update of important parameters. It does not consider the potential positive impact of new tasks on old tasks, which differs from CaGA, which encourages beneficial parameter updates.
>
>
> ## 3. Q1: A scenario of backward knowledge transfer
>
> In the task incremental learning, when new tasks are correlated to old tasks and the new task is more difficult or contains more knowledge, loading the new task parameters can facilitate knowledge transfer to the simpler old tasks [1,2]. For example, given a large language model and an article, the model needs to read the article to perform reading comprehension tasks (question-answering, QA).
>
> The tasks are divided into two types: one is a binary-choice QA, where the model selects one answer from two options; the other is a multiple-choice QA, where the model must choose the correct answer from multiple options (at least four choices), and some options may be similar. The multiple-choice QA is more difficult than the binary-choice QA. Following the normal task sequence, the model first trains and tests on the binary-choice QA, then trains on the multiple-choice QA.
>
> Since the multiple-choice QA is more challenging, requiring more complex reasoning and deeper understanding, after training on it, the model's comprehension of the article will be more profound, and its reasoning abilities will improve. This enhanced ability not only helps the model perform better on the multiple-choice QA but also improves its accuracy on the binary-choice QA, particularly in error correction and fine-grained judgment. In this way, the knowledge and skills learned by the model while handling more complex tasks can be effectively transferred to simpler tasks, ultimately optimizing its overall performance.
>
>
> ## 4. Q3: Generalization on the randomly initialized model
>
> Most existing parameter-efficient fine-tuning methods for incremental learning(e.g., InfLoRA, SAPT, SD-LoRA, etc.),  do not consider validation on randomly initialized models. This is because the randomly initialized model lacks pre-trained knowledge and remains frozen during training, with only a small number of parameters being trained. As a result, it fails to learn effective representations.
>
> To demonstrate this, we evaluate the performance of our method and several baselines on the ImageNet-R benchmark under the 10-task setting using a randomly initialized ViT-B/16 model. In addition to the four metrics reported in the paper, we compute AVG (Top-1), the average of the best Top-1 accuracy across tasks. As shown in Table 2, both our method and the baselines perform poorly with randomly initialized models.
>
> The average of the best task-wise Top-1 accuracy remains just above 10%. These results indicate that parameter-efficient incremental learning frameworks rely on pre-trained models to achieve competitive performance with minimal tunable parameters, supporting the core assumption of parameter-efficient fine-tuning.
>
>
> Table 2: Overall results on ImageNet-R (10 tasks) with the randomly initialized ViT model.
> | Method        | AP↑  | F.Ra↓ | FWT↑  | BWT↑   | AVG(Top 1)↑ |
> | ------------- | ---- | ----- | ----- | ------ | ----------- |
> | SeqLoRA       | 3.42 | 17.57 | -5.86 | -37.43 | 10.68       |
> | InfLoRA       | 3.47 | 17.11 | -6.96 | -34.16 | 10.39       |
> | SD-LoRA       | 4.91 | 15.32 | -4.55 | -33.07 | 12.21       |
> | CaLoRA (Ours) | 5.88 | 15.11 | -4.51 | -31.88 | 12.47       |
>
>
> ## 5. Q5: Comparison with MoE-based methods
>
> Following most existing parameter-efficient fine-tuning methods for incremental learning (e.g., InfLoRA, SAPT, SD-LoRA, HidePrompt, etc.), their comparison methods mainly include prompt-based learning and LoRA-based methods. Thus, our comparison methods are also primarily based on prompt-based learning and LoRA-based methods.
>
> Mixture-of-Experts (MoE)-based continual learning methods introduce multiple expert modules, activating a task-specific subset to minimize interference with prior knowledge. E.g., MoE-P[5] incorporates a Non-linear Residual Gate (NoRGa) mechanism to enhance MoE performance in continual learning. However, compared to LoRA-based lightweight parameter fine-tuning, MoE requires training parameters for multiple experts, which incurs significantly higher overheads in incremental learning scenarios with a large number of tasks.
>
> To highlight the superiority of our method, we compare it with MoE-P under the 10/20 task setting of ImageNetR. As shown in Tables 3 and 4, our method outperforms MoE-P.
>
> Table 3: Comparison results on ImageNet-R (10 tasks)
> | Method        | AP↑       | F.Ra↓    | FWT↑     | BWT↑     |
> | ------------- | --------- | -------- | -------- | -------- |
> | MoE-P         | 73.52     | 6.11     | 1.77     | -7.21    |
> | CaLoRA (Ours) | **77.72** | **2.33** | **4.18** | **0.15** |
>
>
> Table 4: Comparison results on ImageNet-R (20 tasks)
> | Method        | AP↑       | F.Ra↓    | FWT↑     | BWT↑     |
> | ------------- | --------- | -------- | -------- | -------- |
> | MoE-P         | 70.63     | 8.36     | 1.51     | -6.96    |
> | CaLoRA (Ours) | **75.68** | **4.68** | **3.98** | **0.08** |
>
>
>
> [1] Ke, Y., et al. Continual learning of a mixed sequence of similar and dissimilar tasks. *Proceedings of Advances in Neural Information Processing Systems*, 2020, 18493–18504.
>
> [2] Lin, S., et al. Beyond not-forgetting: Continual learning with backward knowledge transfer. *Proceedings of Advances in Neural Information Processing Systems*, 2022, 16165–16177.
>
> [3] Gao, R., et al. Defying catastrophic forgetting via influence function. *Artificial Intelligence*, 339, 104261, 2025.
>
> [4] Kirkpatrick, J., et al. Overcoming catastrophic forgetting in neural networks. *Proceedings of the National Academy of Sciences of the United States of America*, 2017.
>
> [5] Le, M., et al. Mixture of Experts Meets Prompt-Based Continual Learning. *Advances in Neural Information Processing Systems*, 2024.

---

> > ### Comment · Reviewer_5jhA · 2025-08-04
> >
> > Thank you to the authors for the thoughtful analysis and clarifications. Overall, I find the message regarding backward transfer to be unclear. The current results seem to primarily emphasize NLP tasks, where task boundaries are often ambiguous and the observed gains are marginal.
> > I would encourage a more distinct discussion of backward transfer effects on vision and language datasets. In particular, results on image datasets—where task boundaries are more clearly defined—would provide a more compelling case.
> >
> > Regarding the techniques, I appreciate the clarification; they are not the focus of my main critique. Overall, I will maintain my original score.

---

> > > ### Author Response · Authors · 2025-08-05
> > > **New response to task boundary and knowledge transfer**
> > >
> > > # 1. Task boundary and knowledge transfer
> > >
> > > Thank you very much for your valuable feedback.
> > >
> > > We begin by reaffirming a fundamental assumption in task-incremental learning: knowledge transfer is only possible when tasks are correlated. This assumption is well-established not only in incremental learning but also across broader transfer learning frameworks, including multi-task learning [1–10]. In contrast, when task boundaries are clear and tasks are entirely uncorrelated, knowledge transfer does not occur [1–10]. Attempts to transfer knowledge under such conditions contradict this foundational assumption and established understanding in the field [1–10].
> > >
> > > To further support this point, numerous studies have evaluated knowledge transfer using task sequences that include both correlated and uncorrelated tasks [9,10]. E.g., CUBER [9] varifies backward knowledge transfer by constructing 7 overlapping tasks from the first 50 classes of CIFAR-100: Task 1 (0–9), Task 2 (5–14), Task 3 (10–19), Task 4 (20–29), Task 5 (25–34), Task 6 (30–39), and Task 7 (40–49). This design explicitly introduces task correlation through overlapping categories.
> > >
> > > In our response to **W1 & Q4**, we have presented an experiment using NLP tasks with varying degrees of correlation. The results empirically validate the effects of backward transfer. To extend this analysis to the vision domain, we design a special setup using the first 100 classes of ImageNet-R, divided into 6 overlapping tasks: Task 1 (0–19), Task 2 (15–34), Task 3 (35–54), Task 4 (50–69), Task 5 (70–89), and Task 6 (80–99). This design allows controlled investigation of transfer dynamics under different task correlations.
> > >
> > > Table 1 presents the average task correlation and affinity during training, while Table 2 confirms the presence of backward transfer—for example, Task 2 improves Task 1, while Task 4 and Task 6 enhance Tasks 3 and 5, respectively. These results demonstrate that our method accurately estimates task correlation and enables selective backward transfer, thereby mitigating forgetting.
> > >
> > > We have made an extensive effort to address your concerns. We hope our response can effectively address your concerns. If you have any further questions, please do not hesitate to let us know, and we will respond timely.
> > >
> > >
> > >
> > > Table 1: Average task correlation/affinity
> > > | Task  | Task1      | Task2      | Task3      | Task4      | Task 5    |
> > > | ----- | ---------- | ---------- | ---------- | ---------- | --------- |
> > > | Task2 | 0.36/0.27  |            |            |            |           |
> > > | Task3 | 0.13/0.06  | 0.13/0.01  |            |            |           |
> > > | Task4 | 0.11/-0.03 | 0.11/-0.02 | 0.33/0.28  |            |           |
> > > | Task5 | 0.1/-0.03  | 0.13/-0.07 | 0.14/-0.01 | 0.11/-0.03 |           |
> > > | Task6 | 0.13/-0.02 | 0.11/-0.03 | 0.12/-0.01 | 0.13/-0.01 | 0.48/0.25 |
> > >
> > >
> > >
> > > Table 2: Backward knowledge transfer
> > > | Task      | Task1 | Task2     | Task3 | Task4     | Task5 | Task6     |
> > > | --------- | ----- | --------- | ----- | --------- | ----- | --------- |
> > > | Task1 Acc | 91.37 | **91.97** | 91.26 | 90.95     | 90.88 | 90.81     |
> > > | Task2 Acc |       | 92.21     | 92.08 | 91.86     | 91.28 | 91.01     |
> > > | Task3 Acc |       |           | 92.01 | **92.66** | 91.72 | 91.57     |
> > > | Task4 Acc |       |           |       | 92.11     | 91.68 | 91.35     |
> > > | Task5 Acc |       |           |       |           | 91.53 | **92.34** |
> > > | Task6 Acc |       |           |       |           |       | 91.83     |
> > >
> > >
> > >
> > > [1] Efficiently Identifying Task Groupings for Multi-Task Learning. *Advances in Neural Information Processing Systems*, 2021.
> > >
> > > [2] Efficient and Effective Multi-task Grouping via Meta Learning on Task Combinations. *Advances in Neural Information Processing Systems*, 2022.
> > >
> > > [3] Hierarchical Prompt Learning for Multi-Task Learning.  *IEEE/CVF Conference on Computer Vision and Pattern Recognition*, 2023.
> > >
> > > [4] Modeling Task Relationships in Multi-task Learning with Multi-gate Mixture-of-Experts. *ACM SIGKDD International Conference on Knowledge Discovery and Data Mining*, 2018.
> > >
> > > [5] Which Tasks Should Be Learned Together in Multi-task Learning? *International Conference on Machine Learning*, 2020.
> > >
> > > [6] Transfer Vision Patterns for Multi-Task Pixel Learning. *ACM Multimedia Conference*, 2021.
> > >
> > > [7] Understanding the Transferability of Representations via Task‑Relatedness. *Advances in Neural Information Processing Systems*, 2024.
> > >
> > > [8] Efficient Continual Learning with Modular Networks and Task-Driven Priors. *International Conference on Learning Representations*, 2021.
> > >
> > > [9] Beyond not-forgetting: Continual learning with backward knowledge transfer. *Advances in Neural Information Processing Systems*, 2022.
> > >
> > > [10] Continual learning of a mixed sequence of similar and dissimilar tasks. *Advances in Neural Information Processing Systems*, 2020.

---

> > > > ### Comment · Reviewer_5jhA · 2025-08-05
> > > >
> > > > The significance of backward transfer—distinct from general knowledge transfer—requires further justification. Recent advances largely focus on utilizing large pre-trained foundation models that offer broad, transferable knowledge. In contrast, task-to-task transfer typically involves narrower, domain-specific information and limited data, which may be less effective. This raises concerns about the practical utility of the proposed method. Moreover, since all experiments are conducted using a powerful backbone, it becomes more challenging to fairly assess the contribution of backward transfer.
> > > >
> > > > That said, scenarios with partially overlapping classes are indeed realistic. For instance, if the class "cat" appears in both Task 1 and Task 2, improved accuracy on "cat" after learning Task 2 could indicate backward transfer.

---

> > > > > ### Author Response · Authors · 2025-08-07
> > > > > **Further Analysis of Backward Knowledge Transfer**
> > > > >
> > > > > # 2. Further Analysis of Backward Knowledge Transfer
> > > > >
> > > > > Thank you for your positive feedback on backward knowledge transfer and your recognition of its applicable scenarios. In response to your questions, we conduct detailed analyses and validations, with all experiments following the same settings as in the previous response (i.e., dividing the first 100 classes of the ImageNetR dataset into six tasks with overlapping classes). The details are elaborated across the following three aspects:
> > > > >
> > > > > ## 2.1 Pre-trained Knowledge from Large Foundation Models Struggles to Transfer Across Incremental Tasks, and a Powerful Backbone Does Not Undermine the Fairness of Backward Knowledge Transfer Evaluation
> > > > >
> > > > > Although pre-trained models provide powerful pre-trained knowledge, they often lack the fine-grained information required for specific tasks. There is often a distribution shift between the target tasks and the pre-trained knowledge, and fine-tuning for each task is still required for fine-grained knowledge adaptation and correction [1-3]. This discrepancy is even more pronounced in task-incremental learning, where relying solely on pre-trained models cannot achieve optimal incremental learning performance.
> > > > >
> > > > > To validate this, we conduct an experiment on the ViT model without fine-tuning, that is, directly using the pre-trained knowledge of the ViT-B/16 model for incremental inference. As shown in **Table 1**, relying solely on the non-fine-tuned ViT-B/16 model for inference on six tasks yields extremely poor results. This indicates that although pre-trained models possess powerful prior knowledge, they face challenges of poor generalization and difficulty in transfer in challenging incremental scenarios. Relying solely on pre-trained model knowledge cannot achieve effective task performance, let alone effective knowledge transfer.
> > > > >
> > > > > Therefore, pre-trained models with powerful backbone networks do not undermine the fairness of backward knowledge transfer evaluation because relying solely on pre-trained knowledge makes it difficult to achieve knowledge transfer between incremental tasks, including forward and backward knowledge transfer.
> > > > >
> > > > >
> > > > > **Table 1: The results of ViT-B/16 without fine-tuning**
> > > > > | Task       | Task 1 | Task 2 | Task 3 | Task 4 | Task 5 | Task 6 |
> > > > > | ---------- | ------ | ------ | ------ | ------ | ------ | ------ |
> > > > > | Task 1 Acc | 7.8    | 4.72   | 3.59   | 2.78   | 2.46   | 2.13   |
> > > > > | Task 2 Acc |        | 6.75   | 4.22   | 3.3    | 2.61   | 2.38   |
> > > > > | Task 3 Acc |        |        | 3.07   | 2.24   | 2.04   | 1.21   |
> > > > > | Task 4 Acc |        |        |        | 3.91   | 2.46   | 2.41   |
> > > > > | Task 5 Acc |        |        |        |        | 1.98   | 1.82   |
> > > > > | Task 6 Acc |        |        |        |        |        | 2.21   |
> > > > >
> > > > >
> > > > >
> > > > > ## 2.2 Fine-tuning with Limited Task-specific Data Improves Model Knowledge and Facilitates Knowledge Transfer
> > > > >
> > > > > Fine-tuning using limited task-specific data allows the model to retain foundational knowledge while learning task-specific features, providing a necessary foundation for knowledge transfer between tasks [4-5]. To illustrate this, we conduct additional fine-tuning experiments. As shown in **Table 2**, fine-tuning the attention modules of the ViT model significantly improves performance compared to the non-fine-tuned baseline in **Table 1**.
> > > > >
> > > > > Additionally, when Task 2, Task 4, and Task 6 are trained and evaluated independently (i.e., without leveraging knowledge from previous tasks), the accuracies are only 66.24%, 48.57%, and 41.26%, respectively.  However, in incremental scenarios, their accuracies improve to 91.56, 80.58, and 78.95, respectively, because they leverage knowledge from correlated old tasks.
> > > > >
> > > > > This confirms that even limited fine-tuning data can enhance model generalization and promote effective knowledge transfer across tasks.
> > > > >
> > > > >
> > > > >
> > > > > **Table 2: The results of fine-tuning the attention modules in ViT-B/16**
> > > > > | Task       | Task 1 | Task 2 | Task 3 | Task 4 | Task 5 | Task 6 |
> > > > > | ---------- | ------ | ------ | ------ | ------ | ------ | ------ |
> > > > > | Task 1 Acc | 89.81  | 44.01  | 33.01  | 18.28  | 17.64  | 17.61  |
> > > > > | Task 2 Acc |        | 91.56  | 77.47  | 61.61  | 44.14  | 38.62  |
> > > > > | Task 3 Acc |        |        | 81.99  | 56.31  | 47.41  | 29.81  |
> > > > > | Task 4 Acc |        |        |        | 80.58  | 57.77  | 29.13  |
> > > > > | Task 5 Acc |        |        |        |        | 79.9   | 51.4   |
> > > > > | Task 6 Acc |        |        |        |        |        | 78.95  |

---

> > > > > > ### Author Response · Authors · 2025-08-07
> > > > > > **Further Analysis of Backward Knowledge Transfer**
> > > > > >
> > > > > > ## 2.3 Our Method Can Achieve Effective Backward Knowledge Transfer Under Conditions of Limited Data and Narrow Task Boundaries
> > > > > >
> > > > > > While **Table 2** shows that fine-tuning improves task performance, it still suffers from catastrophic forgetting [6]. Existing methods typically address this by restricting parameter updates during training on new tasks to avoid interference with old tasks. For example, InfLoRA [6] mitigates forgetting by constraining updates to be orthogonal to the directions of old tasks.
> > > > > >
> > > > > > However, this method limits the potential for backward knowledge transfer. Recent studies show that allowing selective parameter updates for new tasks can improve the performance of correlated old tasks, thereby enabling backward knowledge transfer and mitigating catastrophic forgetting [7-8].
> > > > > >
> > > > > > To further demonstrate that our method can achieve effective backward knowledge transfer under conditions of limited fine-tuning data and narrow task boundaries, we conduct a comparative experiment. In our settings, the average number of training samples per class is approximately 100, with each task containing only 20 classes. Moreover, there is significant class overlap between Task 1 and Task 2, Task 3 and Task 4,  Task 5 and Task 6, with many similar classes (i.e., narrow task boundaries).
> > > > > >
> > > > > > In contrast, the pre-trained model uses pre-training data from 21,000 classes, totaling approximately 14 million images. Therefore, this setup clearly satisfies the conditions of limited fine-tuning data and narrow task boundaries compared to the vast scale of pre-training data.
> > > > > >
> > > > > > **Table 3** shows the results of InfLoRA. Compared to the fine-tuning results in **Table 2**, InfLoRA better alleviates catastrophic forgetting but cannot achieve backward knowledge transfer between correlated tasks. In contrast, our method can achieve backward knowledge transfer between correlated tasks while maintaining optimal overall performance (as demonstrated in the results presented earlier response under **1. Task Boundary and Knowledge Transfer**).
> > > > > >
> > > > > > Furthermore, to further demonstrate the effect of backward knowledge transfer in our method, **Table 4** shows the performance improvements of our method for five overlapping classes from Task 2 to Task 1. The backward knowledge transfer effect of Task 2 on Task 1 is mainly reflected in the five overlapping classes, with test accuracies for all five classes showing some degree of improvement.
> > > > > >
> > > > > > These results show that our method, by selectively updating parameters beneficial to both new and old tasks, not only preserves performance on old tasks but also achieves meaningful backward knowledge transfer.
> > > > > >
> > > > > > **In summary, under conditions of limited fine-tuning data and narrow task boundaries for pre-trained models, our method achieves backward knowledge transfer by improving the parameter-efficient fine-tuning-based incremental learning paradigm, thereby enhancing the overall performance of incremental learning.**
> > > > > >
> > > > > > *We have made comprehensive efforts to address your questions through detailed explanations and experiments. We hope our responses clarify your concerns effectively. Should you have any further questions or require additional clarification, please do not hesitate to let us know. We would be glad to respond promptly.*
> > > > > >
> > > > > >
> > > > > >
> > > > > > **Table 3: The results of InfLoRA**
> > > > > > | Task       | Task 1 | Task 2 | Task 3 | Task 4 | Task 5 | Task 6 |
> > > > > > | ---------- | ------ | ------ | ------ | ------ | ------ | ------ |
> > > > > > | Task 1 Acc | 90.91  | 89.82  | 87.61  | 85.85  | 84.92  | 84.55  |
> > > > > > | Task 2 Acc |        | 92.14  | 91.03  | 89.28  | 88.32  | 88.31  |
> > > > > > | Task 3 Acc |        |        | 89.51  | 87.4   | 84.88  | 84.81  |
> > > > > > | Task 4 Acc |        |        |        | 88.13  | 84.12  | 82.87  |
> > > > > > | Task 5 Acc |        |        |        |        | 87.5   | 83.25  |
> > > > > > | Task 6 Acc |        |        |        |        |        | 85.97  |
> > > > > >
> > > > > >
> > > > > >
> > > > > > **Table 4: The backward knowledge transfer of 5 classes from task 2 to task 1**
> > > > > > | Class            | Class 1 | Class 2 | Class 3 | Class 4 | Class 5 |
> > > > > > | ---------------- | ------- | ------- | ------- | ------- | ------- |
> > > > > > | Improvement（%） | 2.1     | 2.3     | 2.5     | 2.7     | 2.4     |
> > > > > >
> > > > > >
> > > > > > [1] Surgical fine-tuning improves adaptation to distribution shifts. *ICLR*, 2023.
> > > > > >
> > > > > > [2] When prompt-based incremental learning does not meet strong pretraining.  *CVPR*, 2023.
> > > > > >
> > > > > > [3] An empirical analysis of forgetting in pre-trained models with incremental low-rank updates. *CoLLAs*, 2024.
> > > > > >
> > > > > > [4] Recall and learn: Fine-tuning deep pretrained language models with less forgetting. *EMNLP*, 2020.
> > > > > >
> > > > > > [5] Class-incremental learning by knowledge distillation with adaptive feature consolidation. *CVPR*, 2022.
> > > > > >
> > > > > > [6] InfLoRA: Interference-Free Low-Rank Adaptation for Continual Learning. *CVPR*, 2024.
> > > > > >
> > > > > > [7] Beyond not-forgetting: Continual learning with backward knowledge transfer. *NeurIPS*, 2022.
> > > > > >
> > > > > > [8] Continual learning of a mixed sequence of similar and dissimilar tasks. *NeurIPS*, 2020.

---

### Official Review · Reviewer_TJb6 · 2025-07-05

**Clarity:** 4
**Significance:** 3
**Originality:** 3
**Rating:** 5
**Confidence:** 3

**Summary:**

The paper proposes CaLoRA, a causal-aware low-rank adaptation method for parameter-efficient continual learning that, unlike prior work, explicitly aims to enable backward knowledge transfer from new tasks to old ones. CaLoRA combines parameter-level counterfactual attribution (PaCA) to estimate causal importance of parameters and cross-task gradient adaptation (CaGA) to assess task correlation and affinity. These components adapt gradients during training to selectively update parameters that benefit both new and old tasks. Experiments on NLP and vision benchmarks show that CaLoRA outperforms several strong baselines.

**Questions:**

**Q1.** In Algorithm 1, the causal effect $E_t$ is estimated once at the beginning of training for task t. Have you considered the possibility of updating this causal effect estimation periodically during the training of task t? One might hypothesize that the importance of certain parameters could evolve as learning progresses on the new task.

**Q2.** Can you provide quantitative evidence that gradient similarity is a reliable indicator of positive transfer?

**Q3.** How does the method perform if old task gradients are noisy or approximate?

**Ethical Concerns:**

["NO or VERY MINOR ethics concerns only"]

**Final Justification:**

Thank you for your effort for rebuttal. I keep my score.

**Limitations:**

This paper addressed limitations well, but not addressed potential negative societal impact.

**Paper Formatting Concerns:**

Nothing.

**Quality:**

4

**Strengths And Weaknesses:**

**S1.** This paper tackles the clear motivated problem of backward knowledge transfer in PEFT.

**S2.** The authors use a diverse set of benchmarks from both NLP and Computer Vision, employ modern and large-scale pre-trained models (T5, LLaMA-2, ViT), and compare against a comprehensive list of recent and relevant baselines. The testing across different model scales, task orders, and task lengths demonstrates the method's robustness.

**S3.** CaLoRA consistently outperforms all baselines across all reported metrics. Crucially, it shows substantial improvements in Backward Transfer (BWT), often turning the negative BWT of other methods into a positive value, which directly validates the paper's central hypothesis and contribution.

**W1.** As acknowledged by the authors, CaLoRA requires storing the gradients for all previous tasks, which could lead to significant memory overhead in scenarios with a very large number of sequential tasks. Furthermore, the per-step computation of SVD, gradient projections, and similarities for all previous tasks appears computationally intensive, and a more detailed analysis of this overhead would be beneficial.

**W2.** The method operates in a standard continual learning setting where task boundaries are clearly defined. Its applicability to more complex scenarios like online continual learning, where task boundaries are blurred, is a limitation, as noted by the authors.

---

> ### Author Rebuttal · Authors · 2025-07-31
>
> # Response to Reviewer TJb6
>
> Thank you very much for your detailed review and insightful comments. Below are our responses to your valuable suggestions. We will refine the paper accordingly.
>
> ## 1. W1: Computational overhead analysis
>
> Our method includes PaCA and CaGA modules. Given a pretrained weight matrix $W \in \mathbb{R}^{d_O \times d_I}$, PaCA estimates causal effects via a second-order Taylor expansion with a diagonal Hessian approximation. Both the first-order and second-order terms have a time complexity of $\mathcal{O}(d_O \times d_I)$, making the overall complexity of PaCA linear and comparable to a standard backward pass. This computation can be efficiently parallelized and seamlessly integrated into the training process.
>
> In a task sequence of $T$ tasks, for the current task $T_t$, CaGA performs gradient projection and computes task affinity. The bases of old task gradients are derived through a single singular value decomposition (SVD) and subsequently stored in memory, thereby reducing the memory overhead. Each projection and affinity computation costs $\mathcal{O}(d_O \times d_I)$, leading to a total overhead of $\mathcal{O}((t-1) \times d_O \times d_I)$. Thus, the per-task complexity is approximately $\mathcal{O}(t \times d_O \times d_I)$. The total complexity for the entire sequence is $\mathcal{O}\left(\sum_{t=1}^{T} t \times d_O \times d_I\right)$.
>
> Table 1 shows the time complexity and average per-epoch runtime across 10 tasks on ImageNet-R for our method compared to two LoRA-based baselines. Our method maintains training efficiency on par with projection-based LoRA fine-tuning (i.e., InfLoRA), with only marginal extra cost.
>
> In addition, the previously submitted appendix provides a comparison of memory overhead, showing that our method does not significantly increase storage costs compared to the baselines. Moving forward, we will consider further reducing memory overhead in incremental learning by combining causal effects with techniques such as parameter pruning.
>
> Table 1: Comparison of computational complexity and average per-epoch runtime across 10 tasks on ImageNet-R. $𝑟$ is the rank of LoRA.
> | Method        | GPU Time/Epoch (minutes) | Computational Complexity                                     |
> | ------------- | ------------------------ | ------------------------------------------------------------ |
> | SD-LoRA       | 0.336                    | $\mathcal{O}\left(10 \times r \times d_O \times d_I\right)$  |
> | InfLoRA       | 0.342                    | $\mathcal{O}\left(\sum_{t=1}^{10} t \times d_O \times d_I\right)$ |
> | CaLoRA (ours) | 0.353                    | $\mathcal{O}\left(\sum_{t=1}^{10} t \times d_O \times d_I\right)$ |
>
>
> ## 2. W2: Task boundary
>
> To adapt our task-incremental learning framework to online incremental learning, we will explore two future improvements. The first method adds an inference step. Specifically, during the training phase, we retain the prototype representations corresponding to each task. Then, in the inference phase, we determine which task a test sample belongs to by measuring representation similarity, and subsequently load the parameters of the corresponding task to complete the inference.
>
> When the number of tasks is extremely large, to reduce storage and inference costs, we consider grouping correlated tasks, with only one prototype representation retained for each group. During the inference phase, we determine which group of tasks a test sample belongs to by measuring representation similarity and then load the parameters of the corresponding group of tasks to complete the inference.
>
> The second approach constructs a task-agnostic online learning framework by integrating causal effects and parameter perturbation, enabling model parameters across tasks to converge to a shared parameter space via perturbation. Compared to the first method, it reduces storage costs but may compromise performance on certain tasks.
>
>
> ## 3. Q1: Causal effect in Algorithm 1
>
> Algorithm 1 only outlines the flow across the main modules of the framework. Due to the paper's space constraints, the details of updates in different iterations are not elaborated. Only the initialization of the LoRA parameters is performed once at the beginning of training. Other operations, including causal effect estimation, computation of task correlation and affinity, calculation of the gradient correction term, and gradient update, are adaptively performed at each training step. We will provide a more detailed description of the algorithmic process.
>
>
> ## 4. Q2: Quantitative evidence of the role of gradient similarity in facilitating positive transfer
>
> As discussed in the Task Affinity part, previous studies have shown that gradient similarity reflects task relationships, with positive transfer occurring between tasks with similar gradients [1-4]. In addition, ablation experiments, as shown in Table 5 of the paper, validate the effectiveness of gradient similarity (task affinity) in enabling positive transfer. To further demonstrate this, we conduct an additional experiment.
>
> Specifically, we select five tasks in the Long Sequence benchmark, i.e., BoolQA, IMDB, Yelp, DBpedia, and MultiRec. BoolQA and MultiRec are both question-answering tasks, IMDB and Yelp are sentiment analysis tasks, and DBpedia is a topic classification task. Tables 2 and 3 present the average task correlation and task affinity for these 5 tasks in the initial and middle epochs under the order 1 setting. The results show that correlated tasks, such as IMDB and Yelp, and BoolQA and MultiRec, exhibit higher task correlation and affinity.
>
> Moreover, as gradient updates progress, the correlation between positively correlated tasks increases, while that of negatively correlated tasks decreases, demonstrating the effectiveness of gradient correction. Additionally, Table 4 shows the results of backward knowledge transfer for BoolQA and IMDB. For BoolQA and IMDB, tasks that are positively correlated with them, i.e., MultiRec and Yelp, respectively, transfer knowledge that improves their performance. This suggests that our task correlation and affinity (similarity) can capture the task relationships and facilitate positive transfer.
>
>
> Table 2: Average task correlation/affinity (Initial Epoch) under order 1
> | Task          | BoolQA (Old) | IMDB (Old) | Yelp (Old) | DBpedia (Old) |
> | ------------- | ------------ | ---------- | ---------- | ------------- |
> | IMDB (New)    | 0.11/0.10    |            |            |               |
> | Yelp (New)    | 0.10/0.05    | 0.27/0.21  |            |               |
> | DBpedia (New) | 0.11/-0.09   | 0.14/0.07  | 0.17/0.05  |               |
> | MultiRC (New) | 0.26/0.18    | 0.14/0.03  | 0.14/-0.12 | 0.18/-0.07    |
>
>
>
> Table 3: Average task correlation/affinity (Middle Epoch) under order 1
> | Task          | BoolQA (Old) | IMDB (Old) | Yelp (Old) | DBpedia (Old) |
> | ------------- | ------------ | ---------- | ---------- | ------------- |
> | IMDB (New)    | 0.13/0.11    |            |            |               |
> | Yelp (New)    | 0.13/0.08    | 0.32/0.25  |            |               |
> | DBpedia (New) | 0.11/-0.04   | 0.17/0.05  | 0.20/0.05  |               |
> | MultiRC (New) | 0.32/0.20    | 0.12/0.07  | 0.11/-0.06 | 0.11/-0.03    |
>
>
>
> Table 4: Backward knowledge transfer for BoolQA and IMDB
> | Task       | BoolQA | RTE   | IMDB  | Yelp      | Amazon | SST-2 | DBpedia | AG News | MultiRC   | Yahoo |
> | ---------- | ------ | ----- | ----- | --------- | ------ | ----- | ------- | ------- | --------- | ----- |
> | BoolQA Acc | 86.36  | 86.22 | 86.18 | 86.06     | 86.08  | 86.1  | 86.12   | 86.16   | **86.88** | 86.39 |
> | IMDB Acc   |        |       | 95.87 | **95.96** | 95.96  | 96.05 | 95.98   | 95.94   | 95.52     | 95.9  |
>
>
>
> [1] Yu, T., et al. Gradient Surgery for Multi-Task Learning. *Advances in Neural Information Processing Systems*, 2020.
>
> [2] Liu, B., et al. Conflict-Averse Gradient Descent for Multi-task Learning. *Advances in Neural Information Processing Systems*, 2021, 18878–18890.
>
> [3] Chen, Z., et al. Just Pick a Sign: Optimizing Deep Multitask Models with Gradient Sign Dropout. *Advances in Neural Information Processing Systems*, 2020, 2039–2050.
>
> [4] Lin, S., et al. Beyond Not-Forgetting: Continual Learning with Backward Knowledge Transfer. *Advances in Neural Information Processing Systems*, 2022.
>
>
>
> ## 5. Q3: Performance with noisy old task gradients
>
> We conduct validation experiments on the ImageNet-R benchmark (10 tasks) by adding Gaussian noise to the old task gradients. Since LoRA-based parameter fine-tuning methods involve relatively few parameters, resulting in smaller gradient magnitudes, we test two noise ratios, 0.01 and 0.1.
>
> Tables 5 and 6 present the performance of our method in comparison with InfLoRA, another gradient projection-based method, under the two noise settings. These results show that, even with noisy gradients, our method outperforms InfLoRA. This is likely due to our method's ability to identify effective parameters through causal effects, which helps mitigate the impact of parameter noise to some extent.
>
>
> Table 5: Overall results on ImageNet-R (10 tasks)  with a noise ratio of 0.01
> | Method        | AP↑       | F.Ra↓    | FWT↑     | BWT↑     |
> | ------------- | --------- | -------- | -------- | -------- |
> | InfLoRA       | 72.31     | 4.44     | 0.12     | -5.12    |
> | CaLoRA (Ours) | **77.48** | **2.41** | **4.01** | **0.09** |
>
>
>
> Table 6: Overall results on ImageNet-R (10 tasks) with a noise ratio of 0.1
> | Method        | AP↑       | F.Ra↓    | FWT↑     | BWT↑      |
> | ------------- | --------- | -------- | -------- | --------- |
> | InfLoRA       | 70.12     | 5.78     | 0.05     | -6.07     |
> | CaLoRA (Ours) | **76.51** | **2.97** | **3.78** | **-0.04** |

---

### Comment · Area_Chair_BCT1 · 2025-08-04
**Reminder: Author–Reviewer Discussion**

Dear reviewers,

A friendly reminder that the author–reviewer discussion period will close at August 6, 11:59 pm, AoE. The current mixed ratings on this submission make your final justification particularly valuable. Please engage with the authors’ questions and comments and update your Final Justification accordingly.

Thank you for your time and engagement.

Best regards,

AC

---

### Note · Authors · 2025-08-15

Dear Reviewers, ACs, SACs, and PCs,

We sincerely thank all reviewers for their valuable feedback and are glad our contributions were well received. We appreciate your recognition of our work:

1. Clear motivation and problem formulation of backward transfer (Reviews TJb6, 5jhA, J5Ty), a crucial direction in continual learning with pre-trained models for scalable deployment (Reviews NXah, J5Ty).
2. Novel method using PaCA and CaGA for continual fine-tuning (Review NXah), with thorough ablations validating each component’s effectiveness (Reviews NXah, J5Ty).
3. Comprehensive experiments on NLP and CV benchmarks across various task orders, model scales, and task lengths show that CaLoRA consistently outperforms recent strong baselines (Reviews TJb6, NXah, 5jhA, J5Ty).
4. Substantial backward transfer (BWT) gains, often reversing negative transfer, strongly support the paper’s core contribution (Review TJb6).
5. Excellent presentation with clear writing and good structure (Reviews 5jhA, J5Ty).

We have addressed most of the reviewers' concerns, except for two new questions raised by Reviews 5jhA and NXah. Our new responses are summarized as follows:

1. **Effectiveness and Fairness of Backward Transfer**

   > Experiments demonstrate that the evaluation is fair, using pre-trained models as a consistent basis, and validate the effectiveness of our method in achieving backward transfer under limited data and ambiguous task boundaries.

2. **Can Gradient Projection and Similarity Capture Complex Task Relationships?**

   > Many prior works have established gradient projection and similarity as theoretically sound and effective for modeling task relationships.

   > Our method captures dynamic task relationships and adaptively updates parameters. Extensive experiments across diverse task types, orders, and lengths validate its effectiveness. Notably, results on NLP tasks with ambiguous boundaries confirm the method's ability to capture real task relationships and facilitate backward transfer in complex scenarios.

   > Thus, our method can effectively model task relationships and enable potential backward transfer.

We will refine the paper based on the comments. Our work offers a novel perspective on mitigating catastrophic forgetting in PEFT and, we believe, merits publication to foster further discussion.

We appreciate your constructive comments and look forward to refining our work accordingly.

Best regards,

The Authors of Paper #16688

---

### Decision · Program_Chairs · 2025-09-17

**Decision:**

Accept (poster)

**Comment:**

This paper proposes a novel framework, CaLoRA, which integrates causal-aware representation learning into LoRA-based continual learning (CL) to enable beneficial backward transfer, an underexplored goal in the CL community. The method uses domain causal graphs to guide interventions that improve not only forward retention but also performance on past tasks when learning new ones. Extensive experiments show improvements in backward transfer and overall accuracy across various NLP and vision benchmarks, suggesting that CaLoRA can complement or extend prior LoRA-based methods in CL.

The reviewers were overall positive, highlighting the originality of targeting backward transfer and the effective use of causal reasoning. However, questions remained about the necessity of causal graphs (vs. simpler alternatives), the scope of improvement across all benchmarks, and the added complexity of causal modeling. The authors addressed these concerns with additional ablations and clarifications, which reviewers found helpful. Given the novelty of the objective, sound technical design, and clear empirical gains, I recommend acceptance.